

# Vertical distributions of N₂O isotopocules in the equatorial stratosphere

Sakae Toyoda[1], Naohiro Yoshida[1,2], Shinji Morimoto[3], Shuji Aoki[3], Takakiyo Nakazawa[3], Satoshi Sugawara[4], Shigeyuki Ishidoya[5], Mitsuo Uematsu[6], Yoichi Inai[3], Fumio Hasebe[7], Chusaku Ikeda[8], Hideyuki Honda[8], Kentaro Ishijima[9]

[1] Department of Chemical Science and Engineering, School of Materials and Chemical Technology, Tokyo Institute of Technology, Yokohama 226-8502, Japan
[2] Earth-Life Science Institute, Tokyo Institute of Technology, Tokyo 152-8550, Japan
[3] Center for Atmospheric and Oceanic Studies, Graduate School of Science, Tohoku University, Sendai 980-8578, Japan
[4] Miyagi University of Education, Sendai 980-0845, Japan
[5] National Institute of Advanced Industrial Science and Technology (AIST), Tsukuba 305-8569, Japan
[6] Atmosphere and Ocean Research Institute (AORI), The University of Tokyo, Kashiwa, 277-8564, Japan
[7] Division of Earth System Science, Graduate School of Environmental Science, Hokkaido University, Sapporo 060-0810, Japan
[8] Institute of Space and Astronautical Sciences (ISAS), Japan Aerospace Exploration Agency (JAXA), Sagamihara 252-5210, Japan
[9] Project Team for HPC Advanced Predictions Using Big Data, Japan Agency for Marine-Earth Science and Technology (JAMSTEC), Yokohama, 236-0001, Japan

*Correspondence to:* Sakae Toyoda (toyoda.s.aa@m.titech.ac.jp)

**Abstract.** Vertical profiles of nitrous oxide (N₂O) and its isotopocules, isotopically substituted molecules, were obtained over the equator at altitudes of 16–30 km. Whole air samples were collected using newly developed balloon-borne compact cryogenic samplers over the Eastern Equatorial Pacific in 2012 and Biak Island, Indonesia in 2015. They were examined in the laboratory using gas chromatography and mass spectrometry. The mixing ratio and isotopocule ratios of N₂O in the equatorial stratosphere showed a weaker vertical gradient than the previously reported profiles in the subtropical and mid-latitude and high-latitude stratosphere. From the relation between the mixing ratio and isotopocule ratios, further distinct characteristics were found over the equator: (1) Observed isotopocule enrichment factors (ε values) in the middle stratosphere (25–30 km) are almost equal to ε values reported from broadband photolysis experiments conducted in the laboratory. (2) ε values in the lower stratosphere (< ca. 25 km) are about half of the experimentally obtained values, being slightly larger than those observed in the mid-latitude and high-latitude lower stratosphere. These results suggest the following. (1) The time scale of horizontal mixing in the tropical middle stratosphere is sufficiently large for in-situ photolysis of N₂O, mainly because of strong upwelling and transport barrier between the tropics and extratropics. (2) The air in the tropical lower stratosphere is exchanged with extratropical air on a time scale that is shorter than that of photochemical decomposition of





$N_2O$. Previously observed $\varepsilon$ values, which are invariably smaller than those of photolysis, can be explained qualitatively using a three-dimensional chemical transport model and using a simple model that assumes mixing of 'aged' tropical air and extratropical air during residual circulation. Results show that isotopocule ratios are useful to examine the stratospheric transport scheme deduced from tracer–tracer relations.

**Keywords.** nitrous oxide, stable isotopes, tropical stratosphere, cryogenic sampling, stratospheric transport, stratospheric photochemistry

## 1 Introduction

Nitrous oxide ($N_2O$) is a potent greenhouse gas and stratospheric ozone depleting substance emitted from

10 various sources on the Earth surface. It is injected into the stratosphere in tropical upwelling regions. In the stratosphere, it is transported meridionally by so-called Brewer–Dobson circulation and is decomposed by photolysis (Eq. 1) and photooxidation (Eq. 2) with approximate shares of 90% and 10%, respectively (Minschwaner et al., 1993).

$$N_2O + h\nu \quad \rightarrow \quad N_2 + O(^1D) \tag{1}$$

$$N_2O + O(^1D) \quad \rightarrow \quad 2NO \tag{2a}$$

$$\rightarrow \quad N_2 + O_2 \tag{2b}$$

To establish a detailed picture of stratospheric circulation, concentrations and isotopic information of trace gases such as $N_2O$ are regarded as useful tools. (Plumb, 2007). Natural abundance ratios of $N_2O$ isotopocules, molecular species that only differ in either the number or position of isotopic substitutions (Coplen, 2011), are

20 useful tracers for elucidating the sources and physicochemical records of $N_2O$ because isotopocule ratios reflect the isotopic compositions of source materials, isotope effects specific to each chemical and physical process relevant to formation, transportation, and decomposition of $N_2O$, and mixing of various air and water masses (Toyoda et al., 2015).

Stratospheric distributions of $N_2O$ isotopocules have been studied using scientific balloons and aircraft. For balloon observations, (a) whole air samples are collected using cryogenic samplers and are evaluated using isotope ratio monitoring mass spectrometry (IRMS) (Kim and Craig, 1993; Rahn and Wahlen, 1997; Röckmann et al., 2001; Toyoda et al., 2001; Toyoda et al., 2004; Kaiser et al., 2006); alternatively, (b) in situ measurements are conducted using Fourier-transform infrared spectrometry (Griffith et al., 2000). Such

observations have revealed vertical profiles up to 35 km altitude. For aircraft observations, whole air samples



are collected using pressurizing pumps, with subsequent evaluation by IRMS (Rahn and Wahlen, 1997; Park et al., 2004; Kaiser et al., 2006). Although these observations are limited to ca. 20 km altitude, vertical profiles can be obtained for horizontally wide areas such as Arctic polar vortexes.

Earlier studies showed that a decrease in the mixing ratio of $N_2O$ with altitude is accompanied by enrichment of isotopocules of $N_2O$ heavier than the major one ($^{14}N^{14}N^{16}O$). The results were consistent with the expected isotopocule fractionation during photolysis, but the apparent degrees of fractionation (isotopocule enrichment factor, $\varepsilon$) differed between those of the lower (< 20 km) and middle (> 20 km) stratosphere. They are always smaller than that obtained from laboratory photolysis experiments. Moreover, the values of $\varepsilon$ for the middle

stratosphere were found to depend on the latitude and season. Such variation has been regarded as a result of photochemical and transport processes (Kaiser et al., 2006), but it has not been examined because of the lack of measurements taken over the equatorial upwelling region.

This study was conducted to ascertain the vertical profiles of $N_2O$ and its isotopocules over the equator and to

examine the factors that control the apparent isotopic fractionation in the stratosphere by comparing results from earlier studies.

## 2 Experiments and model simulation

### 2.1 Whole air sampling over the equator

Stratospheric air samples were collected using balloon-borne compact cryogenic samplers (J-T samplers) (Morimoto et al., 2009). The sampler consists of an evacuated 800 $cm^3$ stainless steel sample flask (SUS304), a cooling device called a Joule–Thomson (J-T) mini cooler, a two-liter high-pressure neon gas cylinder, pneumatic valves, solenoid valves, a 100 $cm^3$ high-pressure $N_2$ gas cylinder for actuation of pneumatic valves, an electronic controller with a GPS receiver, a telemetry transmitter, and batteries. The J-T mini cooler can

produce liquid neon from high-pressure neon gas that is pre-cooled by liquid nitrogen. The liquid neon is used as refrigerant to solidify or liquefy the stratospheric air. Contrasted against the larger sampling system used in our previous observations, which was about 250 kg with 12 sample flasks in a Dewar flask filled with liquid helium, the J-T sampler is ca. 20 kg, with operational and logistic advantages at remote sites such as remote islands or polar regions.





Sampling over the eastern equatorial Pacific (0°N, 105°W–115°W) was conducted on February 4, 5, 7, and 8, 2012 during the KH-12-1 cruise of R/V Hakuho-maru, JAMSTEC as a part of the Equatorial Pacific Ocean and Stratospheric/tropospheric Atmosphere Study Program. For each experiment, a 5–8 L STP of air sample was collected at programmed altitude of 19–29 km. The sampler then descended by parachute. It was later recovered on the sea.

Another sampling campaign was conducted at Biak Island, Indonesia (1°S, 136°E) on February 22, 24, 26, and 28, 2015 as a part of Small-Size Project by ISAS / JAXA (Hasebe et al., submitted). For each experiment, two samplers integrated into a single gondola were launched from the observatory of National Institute of Aeronautics and Space of the Republic of Indonesia (LAPAN). Samples were collected at two altitudes. Therefore, in total, we obtained seven samples: two samples on each of four flights, with one failed sampling. Locations of launching sites are shown in Figure 1. Balloon trajectories are portrayed in Figure S2.

## 2.2 Analysis of mixing ratio and isotopocule ratios

At Tohoku University, the mixing ratio of $N_2O$ was measured using gas chromatography with electron capture detection (GC-ECD) with precision of 1 nmol mol$^{-1}$ (Ishijima et al., 2001). The isotopocule ratios, defined as follows, were measured at Tokyo Institute of Technology using gas chromatography – isotope ratio monitoring mass spectrometry (Toyoda et al., 2004; Toyoda and Yoshida, 2016).

$$\delta X = (R_{sample} - R_{standard}) / R_{standard}, \qquad (3)$$

Therein, $X$ denotes $^{15}N^{\alpha}$, $^{15}N^{\beta}$ or $^{18}O$, and where $R$ denotes $^{14}N^{15}N^{16}O/^{14}N^{14}N^{16}O$, $^{15}N^{14}N^{16}O/^{14}N^{14}N^{16}O$ or $^{14}N^{14}N^{18}O/^{14}N^{14}N^{16}O$ of the sample and standards (Toyoda and Yoshida, 1999). The $\delta$ value is expressed as the permil (‰) deviation relative to atmospheric $N_2$, and Vienna Standard Mean Ocean Water (VSMOW), respectively, for nitrogen and oxygen. In addition to $\delta^{15}N^{\alpha}$ and $\delta^{15}N^{\beta}$, the $\delta$ value for bulk N and $^{15}$N-site preference (SP) are often used as illustrative parameters:

$$\delta^{15}N^{bulk} = (\delta^{15}N^{\alpha} + \delta^{15}N^{\beta}) / 2 \qquad (4)$$

$$SP = \delta^{15}N^{\alpha} - \delta^{15}N^{\beta}. \qquad (5)$$

Duplicate analyses were made for a set of two runs: monitoring of molecular ion for determination of $\delta^{15}N^{bulk}$ and $\delta^{18}O$ and $NO^+$ fragment ion for determination of $\delta^{15}N^{\alpha}$. A 300–400 cm$^3$ STP aliquot of the sample air was introduced into the analytical system from the sample flask in a single run. Typical precisions of the isotopic analyses are < 0.1‰ for $\delta^{15}N^{bulk}$, < 0.2‰ for $\delta^{18}O$, and < 0.5‰ for $\delta^{15}N^{\alpha}$, although they were slightly worse for samples collected at higher altitudes because of the lower $N_2O$ mixing ratio.





To analyze the relation between the $N_2O$ mixing ratio ($[N_2O]$) and isotopocule ratio ($\delta$) in a Rayleigh fractionation scheme (Eq. 6), measured values must be normalized with respect to the values before the air mass enters the stratosphere.

$$(1+ \delta)/(1+ \delta_0) = \{[N_2O]/[N_2O]_0\}^{\varepsilon} \qquad\qquad (6)$$

In Eq. 6, subscript 0 signifies a tropospheric value; $\varepsilon$ is the enrichment factor. Because the tropospheric mixing ratio and isotopocule ratios are known to have secular trends, $[N_2O]_0$ and $\delta_0$ were estimated as follows. First, the age of the measured air mass was estimated based on the mixing ratio of $CO_2$, which was also measured for the same air sample (Engel et al., 2009). Then, the $N_2O$ mixing ratio at the time when the air

mass was in the troposphere was calculated using the estimated age of air and the secular trend of tropospheric mixing ratio observed by the ALE/GAGE/AGAGE project (Prinn et al., 2000). We used AGAGE data from Mace Head (Ireland) to calculate $[N_2O]_0$ for stratospheric air in the tropics (this study) and the Northern Hemisphere (in which our previous observations were conducted) and those from Cape Grim (Tasmania) for the Southern Hemisphere (our previous observation in Antarctica). For calculating $\delta_0$, the

secular trends observed at Hateruma island, Japan (Toyoda et al., 2013) and Cape Grim (Park et al., 2012) were used, respectively, for the tropics/Northern Hemisphere and the Southern Hemisphere.

## 2.3 Simulation using a three-dimensional chemical transport model

To examine the factors controlling the stratospheric distributions of $N_2O$ isotopocules, a numerical simulation

was conducted using the Center for Climate System Research/National Institute for Environmental Studies/Frontier Research Center for Global Change atmospheric general circulation model with chemical reactions (CCSR/NIES/FRCGC ACTM) (Ishijima et al., 2010; Ishijima et al., 2015). Because Ishijima et al. (2015) have already given a detailed description of the $N_2O$ isotopocule model, we briefly explain it here.

The $N_2O$ photolysis rate was calculated for 15 bins from 178 to 200 nm and for 3 bins from 200 to 278 nm using a scheme incorporating the parameterization of Minschwaner et al. (1993) (Akiyoshi et al., 2009) and by a main radiation – photolysis scheme of the ACTM (Sekiguchi and Nakajima, 2008). Fractionation of $N_2O$ isotopocules was simulated using wavelength-dependent and temperature-dependent enrichment factors ($\varepsilon$) for $^{14}N^{15}N^{16}O$ and $^{15}N^{14}N^{16}O$ of von Hessberg et al. (2004) and that of $^{14}N^{14}N^{18}O$ estimated from the relation

between apparent $\varepsilon$ for each isotopocule observed in the stratosphere. The model transport was nudged to ERA-interim reanalysis (Dee et al., 2011) for horizontal winds and temperature at 6-hourly time intervals.





Regarding the photooxidation sink of $N_2O$, the concentration of $O(^1D)$ was calculated online in the ACTM; $\varepsilon$ values were calculated as described by Kaiser et al. (2002a).

The main difference of the simulation settings from those in an earlier report (Ishijima et al., 2015) is that the present study did not optimize the photolytic isotopocule fractionation. Therefore, the kinetic fractionations for $N_2O$ isotopocules were provided by the original calculation results obtained using the model chemistry scheme containing experimentally determined enrichment factors described above. However, surface emissions of the four $N_2O$ isotopocules were optimized in the manner described in an earlier report (Ishijima et al., 2015), with emissions modified to reproduce observed trends (Röckmann and Levin, 2005) and interhemispheric differences (Ishijima et al., 2007) of atmospheric $N_2O$ isotopocule mixing ratios. Consequently, the estimated emissions were used for a forward simulation of four $N_2O$ isotopocules in the atmosphere from the surface to the stratosphere in this study. The emissions and tropospheric values are reasonable (see Supplemental Information) compared to those of past studies (e.g., Toyoda et al., 2013; Toyoda et al., 2015) in terms of the necessary order of precision for analysis of the large vertical profiles in the stratosphere in this study.

## 3 Results and discussion

### 3.1 Vertical profiles of the $N_2O$ mixing ratio and isotopocule ratios over the Equator

In all, 11 samples (4 from the eastern Pacific, 7 at Biak Island) were collected at target altitudes; of them, 10 were measured for $N_2O$ isotopocules. Figure 2 presents vertical profiles of the $N_2O$ mixing ratio, $\delta^{15}N^{bulk}$, SP, and $\delta^{18}O$ observed over the equator. Data from our previous observations over Japan, Sweden, and Antarctica and those from observations by Röckmann et al. (2000) and Kaiser et al. (2006) conducted over India are also shown. The height of the tropical tropopause layer (TTL) was 14−18.5 km (Fueglistaler et al., 2009), whereas the tropopause height was 12–16 km over Japan and 9 or 10 km over Sweden and Antarctica (Table S1). As observed at mid-latitudes and high latitudes, the mixing ratio decreases with height; isotopocule ratios increase with height over the equator. However, the vertical gradient is weaker at lower latitudes. Our observation over the equator shows the weakest gradient. Although  a slight difference in mixing ratio was observed for 20–25 km, the two equatorial profiles obtained at different longitudes over the equator agreed quite well. We combined the two datasets as a single one for further examinations.



**3.2 Correlation between mixing ratio and isotopocule ratios: apparent isotopocule enrichment factors**

In Fig. 3, the normalized $\delta^{15}N^{bulk}$ is shown against the normalized $N_2O$ mixing ratio (Rayleigh plot). The equatorial data for lower altitudes (< 25 km) are on the line defined by the data for lower altitudes (< 22–27 km) over middle latitudes and high latitudes. The linear relation is consistent with isotopocule fractionation

during the decomposition of $N_2O$ in a closed system, although the slope of the line, which corresponds to isotopic enrichment factor ($\varepsilon$), is markedly lower than that obtained by laboratory photolysis experiments (see below). However, the three data points obtained at altitudes higher than 25 km show systematic deviation from the line and seem to define another line (Fig. 3b). A similar deviation or *bending* structure of the Rayleigh plot has also been observed at middle to high latitudes (Fig. 3a, from the points where the $x$ axis

value is ca. 0.5) (Toyoda et al., 2004). We therefore compare the slope of the lines obtained for observations at various latitudes and for laboratory simulation experiments.

As portrayed in Fig. 4, absolute values of $\varepsilon$ ($|\varepsilon|$) for $^{15}N^{bulk}$, $^{15}N^{\alpha}$, and $^{18}O$ in the equatorial lower stratosphere are slightly higher than those of middle latitude and high latitude lower stratosphere, but they are still only

about half of the $\varepsilon$ obtained by broadband photolysis experiments (Kaiser et al., 2002b; Kaiser et al., 2003). In contrast, $|\varepsilon|$ in the higher region (or middle stratosphere) show larger values. They are the largest over the equator except for $\varepsilon(^{18}O)$. The equatorial values of $\varepsilon$ almost coincide with those of photolysis. It is also noteworthy that latitudinal and year-to-year or seasonal variation are slight compared to those of the middle stratosphere in the lower stratosphere.

**3.3 Cause of the variation of stratospheric $\varepsilon$**

We then discuss causes of (1) lower $|\varepsilon|$ value in the lower stratosphere, (2) increase of $|\varepsilon|$ in the middle stratosphere, and (3) the largest $|\varepsilon|$ in the equatorial middle stratosphere based on two factors: photochemical and transport processes.

**3.3.1 Photochemical processes**

During photochemical decomposition of $N_2O$, $\varepsilon$ reportedly depends on the wavelength that photolyzes $N_2O$, the relative share of photolysis and photooxidation pathways (Eqs. 1 and 2), and temperature (Toyoda et al., 2004; Kaiser et al., 2006). The difference in the ratio of $\varepsilon$ values or normalized $\delta$ values for independent isotopocules (e.g., $\varepsilon(^{15}N^{bulk})/\varepsilon(^{18}O)$ or $\delta^{15}N^{bulk}_{norm}/\delta^{18}O_{norm}$) has been identified as a useful parameter to distinguish photolysis and photooxidation (Kaiser et al., 2002a) because its sensitivity to wavelength and

temperature is small. In Fig. 5, we plot the normalized $\delta$ values obtained in this study and some previous ones.


Especially in Fig. 5b, almost all data show a compact relation. There is no bending or curved structure apparent in the Rayleigh diagram, which suggests that chemical processes are not variable, although the small fluctuation in the lower left region in Fig. 5a will be discussed later.

### 3.3.2 Transport processes

5  As for transport, Kaiser et al. (2006) pointed out that in a simple 1D reaction–diffusion scheme, apparent or effective $\varepsilon$ can be reduced from $\varepsilon$ for the intrinsic sink reaction depending on the ratio of time scales for transport and chemistry as follows:

$$\varepsilon_{\mathrm{eff}} \approx \frac{1}{2}\varepsilon_{\mathrm{sink}}\left(1 + \frac{1}{\sqrt{1 + \tau_{\mathrm{trans}}/\tau_{\mathrm{chem}}}}\right) \quad . \tag{7}$$

In Eq. 7, $\tau_{\mathrm{trans}}$ and $\tau_{\mathrm{chem}}$ respectively denote the time scales for transport and chemistry; $\tau_{\mathrm{trans}} = 4H^2/K$ and $\tau_{\mathrm{chem}} = 1/k$, where $H$, $K$, and $k$ respectively represent the scale height, the vertical eddy diffusion constant, and the rate constant of chemical decomposition of $N_2O$. If $N_2O$ decomposition is limited by transportation rather than by photolysis, then ratio $\tau_{\mathrm{trans}}/\tau_{\mathrm{chem}}$ is high. Consequently, $\varepsilon_{\mathrm{eff}}$ approaches to about half of $\varepsilon_{\mathrm{sink}}$ for the photolysis.

15  Conversely, if $N_2O$ decomposition is limited by photolysis rather than transportation, then $\tau_{\mathrm{trans}}/\tau_{\mathrm{chem}}$ is small and $\varepsilon_{\mathrm{eff}}$ approaches to $\varepsilon_{\mathrm{sink}}$. Based on this simple scheme, Kaiser et al. (2006) explained qualitatively the vertical and meridional trends in $\varepsilon$ obtained over latitudes ranging from 18° to 80° (N or S). Results of this study demonstrate that $\varepsilon_{\mathrm{eff}}$ indeed equals to $\varepsilon_{\mathrm{sink}}$ in the equatorial middle stratosphere, indicating that the time scale of photochemistry is sufficiently larger than that of vertical transport.

We then consider the effect of transportation with a conceptual mixing model extending the 1D scheme to 2D, based on a simplified two-dimensional circulation model in the tropical and extra-tropical stratosphere that was proposed to explain tracer–tracer correlation (Plumb, 2002). In the tropics, $N_2O$ is decomposed gradually during upwelling of the air mass injected from the troposphere. The uppermost tropical air mass $X_0$ is then

25  transported to middle latitudes and higher latitudes, where it begins downwelling. Although a barrier exists between the tropics and mid-latitudes and high latitudes, entrainment of air mass $Y_i$ across the subtropical edge separating the two regions must occur to compensate mass flux in the lower region (Fig. 6).

If we assume tropical profiles of $N_2O$ and its isotopocule ratios (e.g., $\delta^{15}N^{\mathrm{bulk}}$) are determined purely by

30  photochemistry with initial mixing ratio of 320 ppb, a delta value of 0‰ and a $\varepsilon_{\mathrm{sink}}$ of −50‰, then tropical air masses vertically divided from $Y_8$ through $Y_1$ and $X_0$ are expected to line up on a solid line as portrayed in Fig. 7. Next, let us consider that air mass $X_0$ is mixed with $Y_1$ to form $X_1$. Based on the mass balance of





isotopocules before and after mixing, the resulting composition of $X_1$ is obtained as a curve, as shown in red in Fig. 7. Assuming arbitrarily that the mixing ratio of $Y_1$ to $X_1$ is 0.1, and repeating such mixing stepwise, then we obtain mixing ratio and isotope ratios of $N_2O$ in $X_1$ through $X_8$ as black stars in Fig. 6. This hypothetical, continuous mixing produces a curve that is qualitatively consistent with observations made over

the mid-latitudes or high latitudes.

The mixing effect must also be the cause of smaller $\varepsilon$ in the equatorial lower stratosphere (Fig. 4). The mean age of air deduced from $CO_2$ mixing ratio is known to be significantly larger than the phase lag of the water vapor mixing ratio, a so-called tape recorder signal, in the tropical stratosphere, which is explainable by

mixing of old air from the extratropics into the tropics (Waugh and Hall, 2002). In addition, the difference in age between the equator and mid-latitude (over Japan) decreases concomitantly with decreasing altitude (Sugawara et al., unpublished data), suggesting that the time scale of meridional mixing or transport is smaller in the lower stratosphere than in the middle stratosphere, as suggested by results of an earlier study (Boering et al., 1996).

### 3.3.3 Comparison with chemical transport model (ACTM) simulation

We further examined the importance of transport using ACTM. Figure 8 presents results of ACTM simulation with observational data. Although the model approximates the photolysis of $N_2O$ in the longer wavelength region ($\lambda > 200$ nm) with lower spectral resolution, profiles of the $N_2O$ mixing ratio and isotopocule ratios

were reproduced well, except in the winter polar stratosphere, where dynamic processes specific to the polar vortex might not be simulated appropriately in the model. In Fig. 9, the model simulation and observations are compared on a Rayleigh plot. Again, the model reproduced the difference between tropical and mid-latitudes or high latitudes. Because in situ $\varepsilon_{\text{photolysis}}$ used in the model calculation is nearly the same between low and high latitudes (Fig. S3a), this agreement supports the inference that the major causes of the

difference are transport and mixing.

### 3.4 Share of photolysis and photooxidation

Kaiser et al. (2006) used the ratio of normalized $\delta^{15}N^{\text{bulk}}$ and $\delta^{18}O$ ($\psi$) and the ratio of normalized $\delta^{15}N^\alpha$ and $\delta^{15}N^\beta$ ($\eta$) to estimate the relative share of photolysis and photooxidation based on the fact that $\psi$ and $\eta$ are

almost independent of transport processes and are significantly different between the two decomposition processes. In Fig. 10, we show $\psi$ and $\eta$ values for data in Fig. 2. Although it is noteworthy that errors in $\psi$





and $\eta$ values increase concomitantly with decreasing altitude because of the decrease in the normalized $\delta$ values, low values are obtained near the TTL over the Equator just as they are at other latitudes. This result confirms the indication by Kaiser et al. (2006) that the photooxidation sink has a much larger fraction than 10% in the lower stratosphere. A rapid increase of the share of photooxidation in the lower stratosphere is

5 also predicted in the ACTM used for this study (Fig. S3b). Because $O(^1D)$ is more abundant in the tropical stratosphere than in the middle latitude and high latitude stratosphere because of stronger solar irradiation, the signal of photooxidation sink can propagate from the tropics to the extratropics by transport.

## 4 Conclusions

Vertical profiles of isotopocule ratios of $N_2O$ in the equatorial stratosphere are found using balloon-borne

compact cryogenic samplers and mass spectrometry in the laboratory. This report of the relevant literature is the first describing observations of them over the equator. Enrichment factors for isotopocules in the middle equatorial stratosphere (25–30 km) agreed with those obtained with laboratory photolysis experiments, suggesting that the isotopocule ratios are determined mainly by photolysis because of weak vertical or horizontal mixing in the tropical upwelling. In the lower equatorial stratosphere (< ca. 25 km), $N_2O$ is likely

to be decomposed by photochemical processes with a larger contribution from photooxidation than that estimated for the whole stratosphere. It is latitudinally well mixed in the lower stratosphere. Distribution of $N_2O$ and its isotopocules in the middle stratosphere depends on mixing during meridional circulation. Further observations of temporal variations and comparison with ACTM simulation will make isotopocules an effective tool for probing of variation in transport processes and decomposition processes.

*Competing interests.* The authors declare that they have no conflict of interest.

*Acknowledgments.* We thank researchers and crew of R/V Hakuho-maru KH-12-1 cruise for sampling over the eastern equatorial Pacific, and researchers and technical staff of LAPAN for the sampling over Biak. This work was supported by JSPS KAKENHI Grant Numbers 23224013 and 26220101 and also by JAXA as its Small-Size Project.

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

Figure captions

**Figure 1: Map showing balloon launching sites.**

**Figure 2: Vertical profiles of mixing ratio (a), $\delta^{15}$N$^{bulk}$ (b), *SP* (c), and $\delta^{18}$O (d), of N$_2$O observed over the equator (pink symbols). Previously published results obtained over Japan (black and red symbols), Sweden (blue), and Antarctica (green) (Toyoda et al., 2001; 2004), and India (orange, Kaiser et al., 2016; Röckmann et al. 2001) are also shown. In the legend, launch sites and dates are shown, respectively, by three characters and six digits in yymmdd format. See also Table S2 for details.**

**Figure 3: Correlation between mixing ratio and $\delta^{15}$N$^{bulk}$ of N$_2$O (Rayleigh plot). The high mixing ratio range (> ca. 120 nmol mol$^{-1}$) in (a) is enlarged in (b). Both parameters are normalized to their values at the time when the corresponding air mass entered the stratosphere (see the text). Grey solid and broken lines show slopes obtained respectively from laboratory broadband photolysis experiments (Kaiser et al., 2002; 2003) and photooxidation experiments (Kaiser et al., 2002; Toyoda et al., 2004).**

**Figure 4: Comparison of absolute values of the isotopocule enrichment factor (|$\varepsilon$|) for $^{15}$N$^{bulk}$, $^{15}$N$^{\alpha}$, and $^{18}$O of N$_2$O between observations and laboratory experiments. L and M respectively refer to the lower and middle stratosphere with boundary altitude of 20–27 km based on the Rayleigh plot shape (Fig. 3). The respective |$\varepsilon$| for Japan, Sweden, and Antarctica are from Toyoda et al. (2004). Those for photolysis and photooxidation experiments are referred from reports by Kaiser et al. (2002; 2003) and Toyoda et al. (2004). Error bars show either the standard deviation for the mean value (observation over Japan and photooxidation experiments), standard error associated with linear regression in Rayleigh plot (observations except Japan), or the possible range for stratospheric conditions (photolysis experiments).**

**Figure 5: Correlations between $\delta^{15}$N$^{\beta}$ and $\delta^{15}$N$^{\alpha}$ of N$_2$O (a) and between $\delta^{18}$O and $\delta^{15}$N$^{bulk}$ of N$_2$O (b). The δ values are normalized as noted in the text. Grey solid and broken lines show slopes obtained respectively from laboratory**




broadband photolysis experiments (Kaiser et al., 2002; 2003) and photooxidation experiments (Kaiser et al., 2002; Toyoda et al., 2004).

Figure 6: Conceptual two-dimensional circulation model to analyze mixing processes between tropics and extratropics (from Plumb, 2002). The $X_0$ is the uppermost tropical stratospheric air mass, $X_i$ ($i = 1–8$) are air masses formed by mixing of $X_{i-1}$ and $Y_i$.

Figure 7: Presentation of $X_i$ (black stars) obtained using the mixing model with assumed $Y_i$ in the Rayleigh plot. The straight line shows tropical vertical isotopocule fractionation without transport/mixing effect. Curves show mixing between $X_{i-1}$ and $Y_i$ where mixing ratio $Y_i/X_i$ is assumed to be 0.1.

Figure 8: Comparison of vertical profiles of mixing ratio (a) and $\delta^{15}N^{bulk}$ (b) of $N_2O$ between observations and simulation by the ACTM. Model simulations for equatorial profiles were conducted for two dates because the observations were conducted during a 5-day or 7-day period.

Figure 9: Comparison of results of ACTM simulation and stratospheric observation in Rayleigh plot. The square region shown by broken lines in panel (a) is enlarged in panel (b).

Figure 10: Vertical profiles of ratio of normalized $\delta^{15}N^{bulk}$ and $\delta^{18}O$ ($\psi$) and the ratio of normalized $\delta^{15}N^{\alpha}$ and $\delta^{15}N^{\beta}$ ($\eta$). Grey bands show values obtained by laboratory broadband photolysis experiments (Kaiser et al., 2002; 2003) and photooxidation experiments (Kaiser et al., 2002; Toyoda et al., 2004) with widths representing their uncertainty.





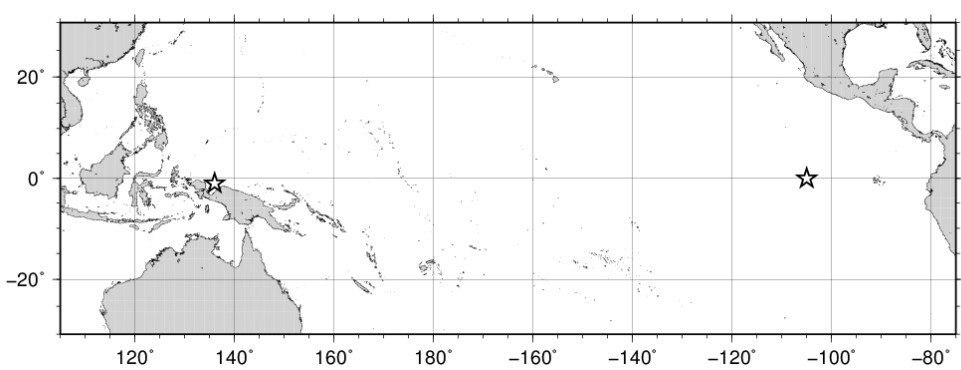

Fig. 1

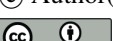



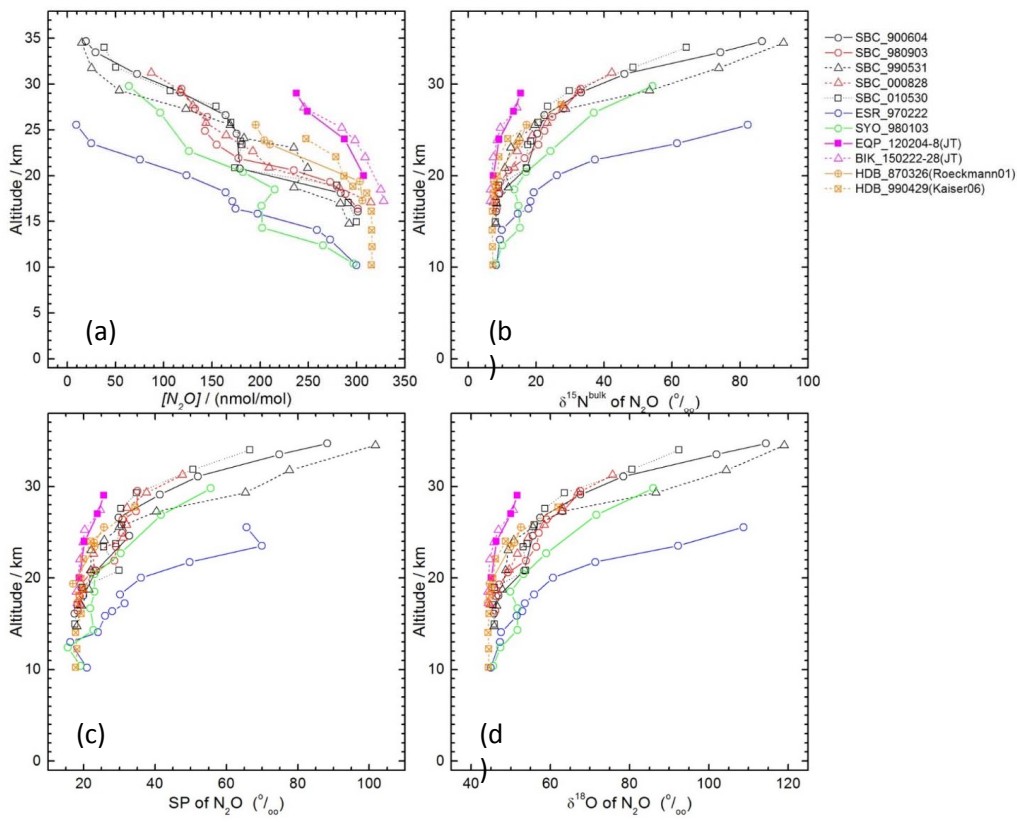

Fig. 2





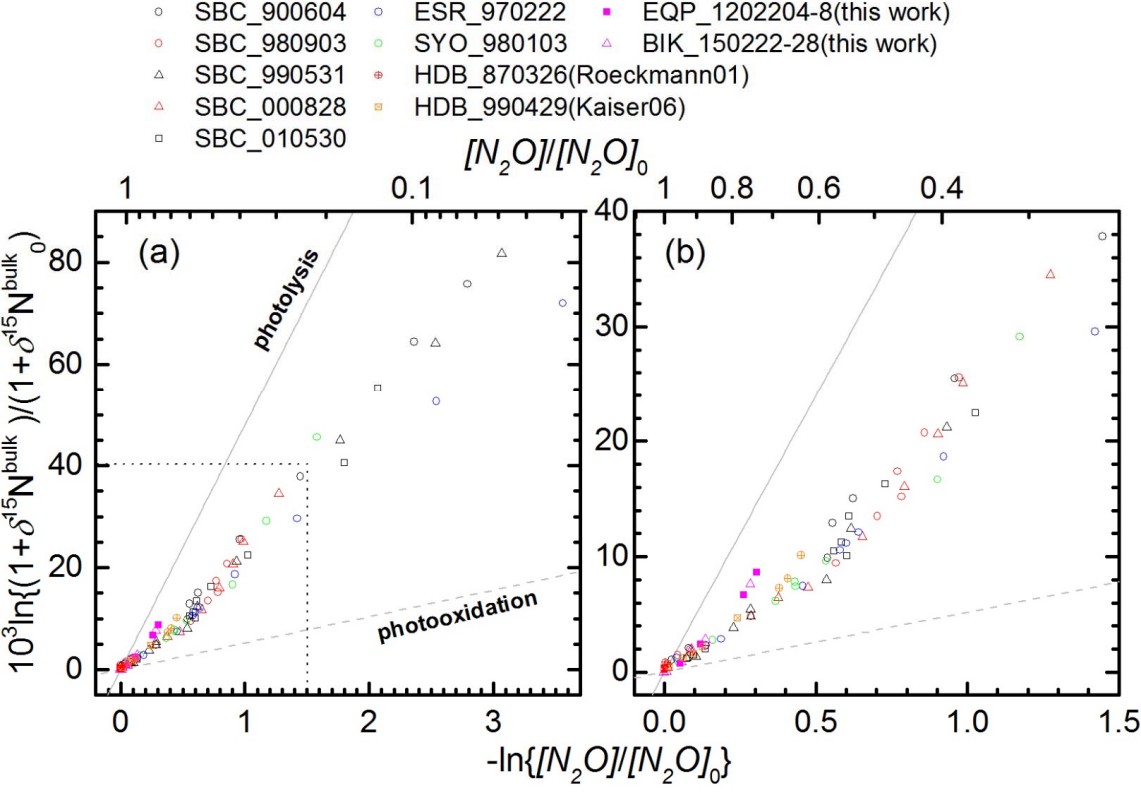



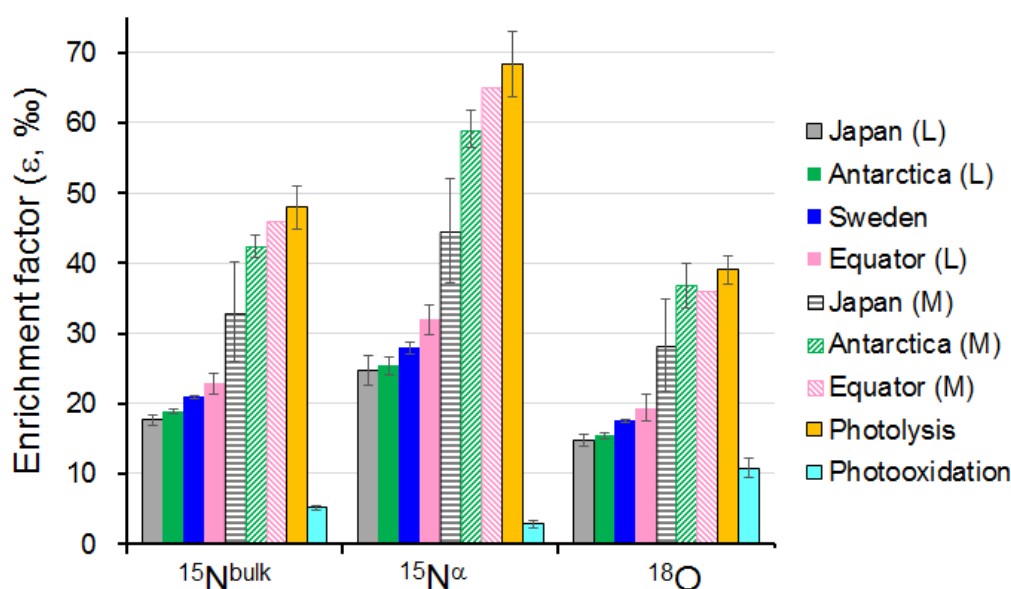

Fig. 4





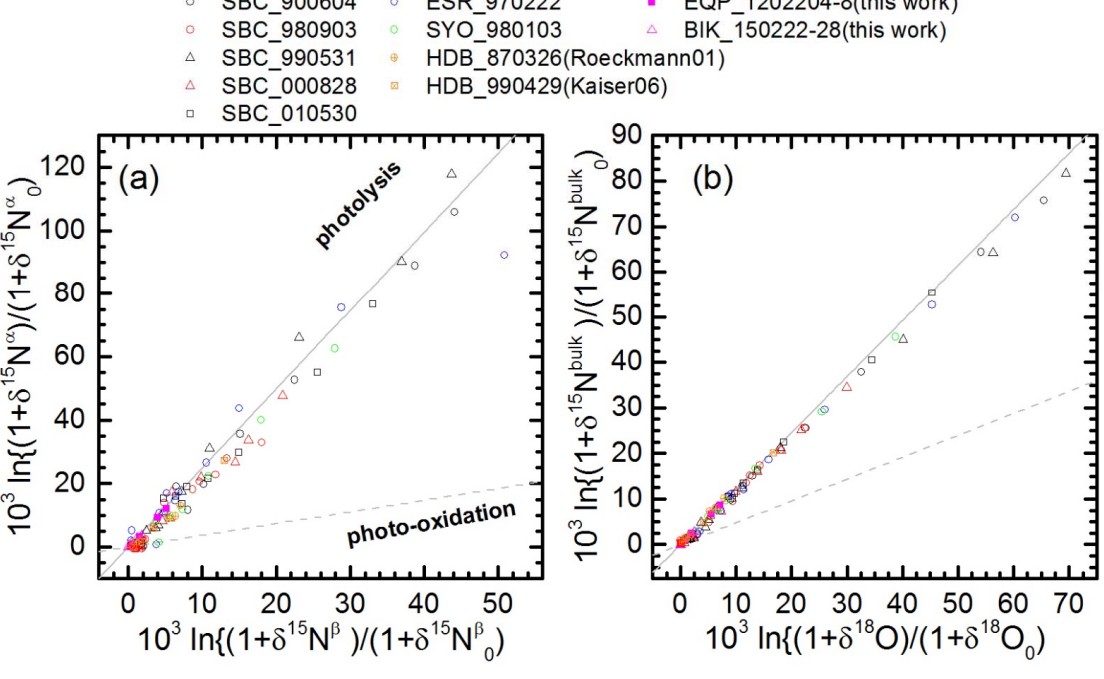

Fig. 5




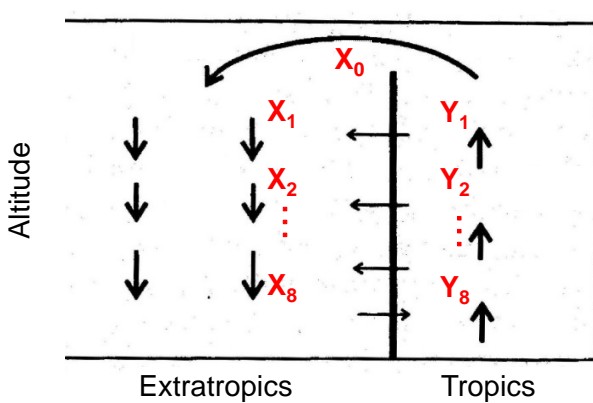

Fig. 6




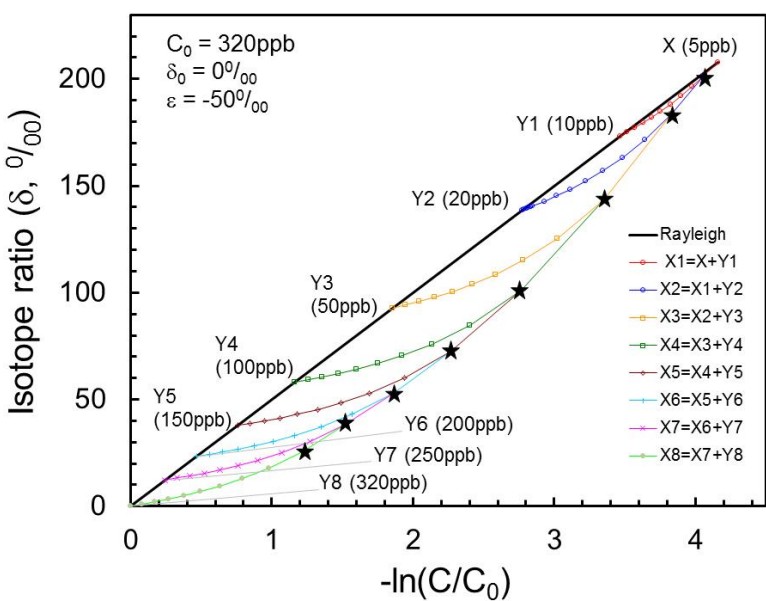





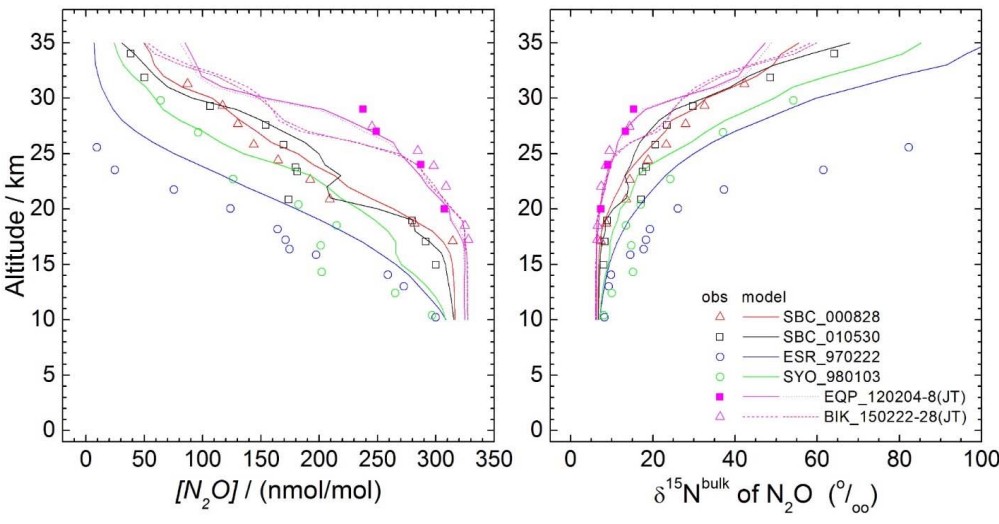

Fig. 8





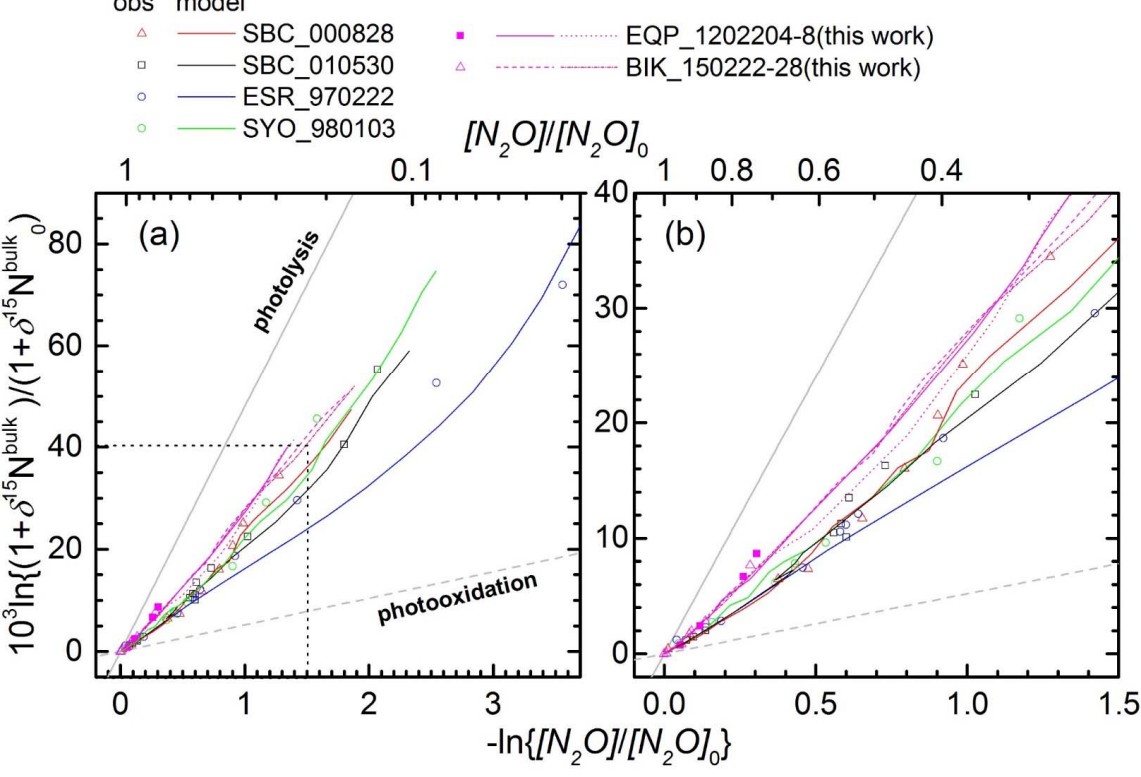





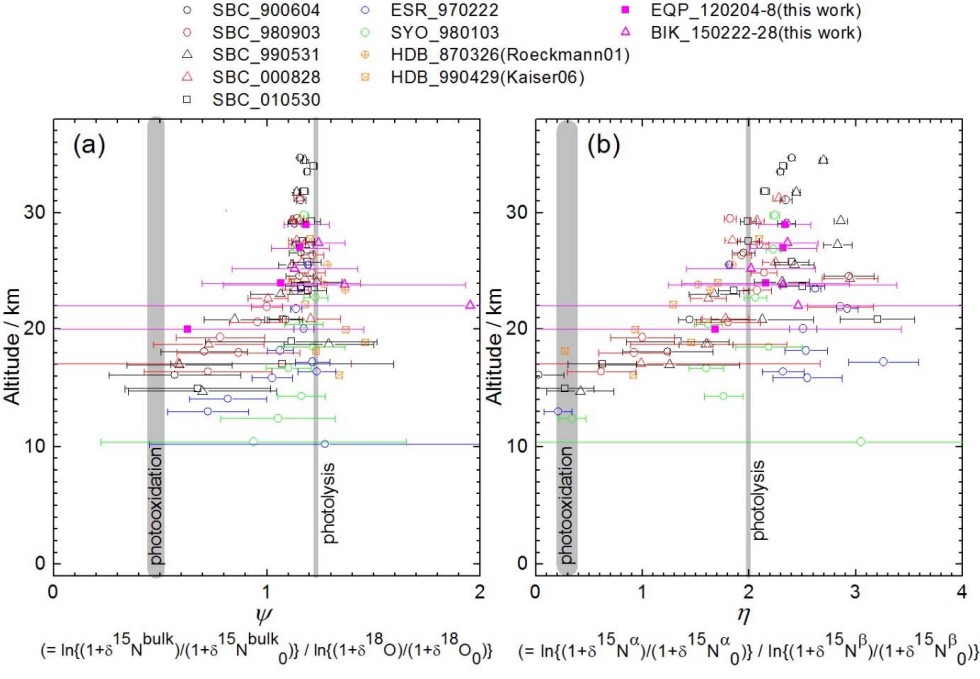

Fig. 10