# Peer review of "Vertical distributions of $N_2O$ isotopocules in the equatorial stratosphere"

_Atmospheric Chemistry and Physics, 2017_

## Referee Comment (RC1) · Anonymous Referee #1 · 3 Jul 2017

This paper extends previous measurements and analysis of the vertical distribution of N2O isotopocules into the tropical stratosphere. It introduces a new sampling system with a much weight-reduced cryogenic sampler. The measurements are impressively accurate and precise and show systematic differences to previously published mid latitude profiles, with implications for the balance between photolysis and photooxidation of N2O, and transport, in determining the vertical distributions. The paper is clearly written and presented. It is well suited to ACP and I recommend publication as is except for a few technical edits as listed below.

Technical comments:

P3 L11: . . . but it has not been FULLY examined because of. . .

[Figure]

P8 L13 and elsewhere – please replace all instances of "transportation" with "transport".

P8 L14 & 15: I suggest using "photochemistry" rather than "photolysis" here because the first reference on L14 is to both photolysis and phot-oxidation.

P8 L19: "faster" would be more unambiguous than "larger" in this context

Caption figure 2 – it would be helpful here to list the site names and three letter abbreviations so the reader is not obliged to go to the supplementary material to find out. Defined once is enough to cover all later figures.

---

## Referee Comment (RC2) · Anonymous Referee #2 · 4 Jul 2017

In this manuscript, Toyoda et al present the first vertical profiles of the isotopic composition of N2O in the deep tropical stratosphere (i.e., quite literally over the equator) that extend above 21 km. They use a new, unique very small cryogenic whole air sampler that can be launched from ships using small balloons and retrieved from the ocean surface after a water landing; this is critical since more traditional cryogenic whole air samplers are extremely heavy and therefore must be launched with very large balloons, greatly limiting their deployment in the deep tropics, especially if they must also be designed to survive ocean landings in addition to the difficulty of finding suitable launch sites near the equator. The magnitudes of the isotopic fractionation of N2O in the deep tropics and the character of the relationships between the N2O isotopic compositions with N2O mixing ratios have been a topic of speculation for over a decade, especially

given the promise that deep tropical data up to ∼30 to 35 km could hold for further elucidation of the patterns and rates of transport between the tropics and extratropics and for the competition between transport and photochemistry in determining the isotopic composition of N2O. This first new deep tropical dataset with measurements up to 30 km is therefore exciting and interesting. These data, their comparison with previous measurements and model results from a number of research groups, and their analysis will therefore be of great interest to the stratospheric and isotope communities of researchers. That said, there are a number of improvements to the analysis, discussion, and presentation of the data that I think should be considered before publication.

Major points to consider:

1. A more clear motivation for the need for and applications of N2O isotope observations in the tropics could be given in the introduction as well as in the discussion and conclusions. The context and questions that remain unanswered or are controversial (or at least that some of us think are controversial and not proven) are not clearly outlined or discussed in the manuscript as currently laid out. In addition, many other tracer measurements have been used to study transport and mixing – what might be unique about N2O isotopocules? This could be mentioned, especially in the concluding section.

2. A lot of detail is given to 'normalize' the N2O isotope and mixing ratio data since there is quite a large time span of datasets with which the new tropical observations are compared. There are also ratios of epsilon values, which in some cases may be referred to as 'normalized' (i.e., relative to the tropopause). It is unclear to me what 'age corrected' and 'normalized' mean in various places. And it is also unclear which data are plotted in the main text and in the supplementary material. All this information should be made more explicit and clear in both the main text and the figure captions. I think it is also important to plot up and/or give in tables both the 'raw' data as well as the age-corrected/normalized data for comparison. I have further specific comments on this topic below.

3. My own opinion is that analyzing the isotope data in subsets of altitude bins (which depend on latitude) is a lot less illuminating than with respect to grouping by N2O mixing ratio, which is in some ways less complicated than altitude given the Brewer-Dobson circulation bringing much older air down from higher altitudes in the extratropics and the polar vortices. Indeed, by binning by N2O mixing ratio, aircraft data from other researchers and I'm sure from the authors' own aircraft datasets can be more easily compared, rather than being limited here by vertical profiles obtained by balloons. This might also be why the Esrange flight from Sweden in early February was not split into a 'lower' and a 'middle' because figuring out where to put the altitude bin might have been difficult for this flight since N2O is so low at a given altitude relative to the other balloon profiles. For example, the data plotted in Figure 3 for the Swedish vortex flight beg to be plotted into 2 groups "M" and "L" (for 'middle' and 'lower' respectively), and yet they are given as one group. It would be interesting to see how small the apparent epsilon value would be for the vortex data, as well as to compare that with remnant vortex data on samples collected by aircraft from Park et al. [2004]. The more complicated the transport and mixing history of air masses in different regions of the stratosphere and/or at different times, the lower the apparent epsilon value is, and the Esrange data and some aircraft data could be very convincing in showing this, and contrasting it with these first deep tropical data at altitudes >~23 km and N2O mixing ratios <~250 ppb, for which transport and mixing have the least influence compared with the rest of the stratosphere. This is the point the authors are trying to make but I think it could be made much more clear.

4. The exciting thing about the new deep tropical dataset is that it clearly shows what we all expected: that in the deep tropics where the stratosphere is effectively the most 'isolated' relative to all other regions, the fractionation for all the N2O isotopocules is as 'Rayleigh' as it gets, and that the apparent epsilon values are close to that for photolysis in the lab. However, rather than discuss this qualitatively from a "Rayleigh" perspective, and using other stratospheric data/observations to show that this is consistent with what we know about relative transport and chemical time scales in different regions

(and seasons) in the stratosphere, the authors instead jump right to an equation from Kaiser et al [2006] (which also appears in Kaiser et al. 2002a, an observation and lab study, and Morgan et al. 2004, a modeling study) that is an approximation to an approximation for a 1D model of the stratosphere. This would, I suppose, be all right except that – given what we know about transport time scales and the photochemical lifetime of N2O in the tropics and the extratropics as a function of altitude from models and from other stratospheric observations (such as N2O, CO2, H2O, CH4 and other long-lived tracers and their correlations) – this 1D model analogy I think actually gives us the wrong answer! Thus, I find this section misleading and not especially useful or illustrative at best and incorrect at worst.

Here is my point as to why I think simple departures from a Rayleigh model based on how much transport and mixing there has been relative to other regions of the stratosphere are more enlightening than this pseudo 1D model treatment: First, below 22 km, transport time scales are fast (and the pseudo 1D treatment convolves both vertical and horizontal/quasi-isentropic transport as 'eddy diffusion') and the photochemical decomposition of N2O is very slow (by both photolysis and reaction with O(1D)). At 20 km, the transport time scale may be months while the e-folding time for N2O destruction is 70 years, for example. If you plug t(trans)«t(chem) into Equation 7, however, the result is epsilon effective = epsilon sink. In contrast, higher up in the tropical stratosphere, t(trans)»t(chem) [where the photochemical destruction really starts picking up speed], so now Equation 7 simplifies to epsilon effective = $\frac{1}{2}$ epsilon sink. But that's exactly where one would expect Rayleigh fractionation to hold – where the change in N2O isotopic composition is going to follow the change in N2O mixing ratio according to epsilon for the sink without transport messing things up. So these two limits from Equation 7 are the opposite of what the new deep tropical observations show! At midlatitudes below 30 km where t(trans)«t(chem) for N2O [again, from many other stratospheric 'non-isotope' studies], Equation 7 would give epsilon effective = epsilon sink and yet the observations show it is more like the other limit, epsilon effective = $\frac{1}{2}$ epsilon sink. So also the opposite of the Equation 7 limits. Indeed, even Kaiser et

al. [2006] say the following: "even though the apparent fraction constants could possibly be reconciled with certain diffusion coefficients and [N2O] destruction rates, the [observed] increase of the magnitude of the apparent fractionation constant with altitude is actually the opposite to what one would expect from a simple vertical reaction-diffiusion-advection model. . . Despite the conceptual usefulness of this 1D description, it has to be stressed that stratospheric transport cannot be characterized as a function of altitude alone and that vertical diffusion is actually not an important process, even though it can be helpful to describe stratospheric transport to some extent. Meridional transport schemes have to be included to explain the variation of epsilon apparent." Thus, Equation 7 gives the opposite answer to all the N2O isotope observations I am aware of, including these new ones in the deep tropics, and even Kaiser et al [2006] had to invoke what we know about meridional patterns of transport and mixing to explain all the stratospheric observations available in 2006. Thus, it is not true, as stated by the authors in this manuscript on Lines 15-20 on Page 8 in this section, that Kaiser et al "used this simplified scheme in Equation to describe qualitatively the vertical and meridional trends in epsilon [values] obtained over latitudes ranging from 18 to 80 (N or S)." Hence I do not find Equation 7 nor the discussion around it to be useful and I think it is actually wrong to use it here. The 2D nature of the Brewer-Dobson circulation and the concomitant chemical and transport timescales associated with the transport and mixing in 2D is required to understand the distribution of N2O isotopes and their relationship with N2O mixing ratios everywhere; Equation 7 does not work.

That said, the 2D mixing scheme the authors use does qualitatively work for me. While I'm not sure where their '10%' number comes from (it's arbitrary), qualitatively it can explain the 2D mixing patterns relative to the photochemical time scales for N2O destruction and why (a) at higher altitudes in the tropics, epsilon apparent = epsilon sink (again a very exciting set of observations) and (b) at lower altitudes in both the tropics and the extratropics, epsilon apparent is more like $\frac{1}{2}$ epsilon sink). Thus, I recommend simply removing the 1D model equation and discussion entirely. It leads the reader to the opposite conclusion, as has been pointed out previously, while the qualitative mixing model is actually more realistic (or at least illustrative), despite it perhaps seeming somewhat arbitrary to some readers.

And the 3D model comparison is a nice follow-on to the 2D results and discussion, and shows quite a nice quantitative agreement with the observations in Figures 8 and 9. In fact, the authors conclude that "Because in situ epsilon photolysis used in the model calculation is nearly the same between low and high latitudes (Fig. S3a), this agreement supports the inference that the major causes of the difference [in epsilon values] are transport and mixing." I agree, and Morgan et al. [2004] have similar results from their 2D CTM results.

The 1D Equation 7 conclusions being the opposite of what these and all past data (I think) show, if the authors modify their approach on this, then these ideas also need to modified in the abstract and conclusions as well, where in some cases the opposite of 'transport' vs 'chemical time scales' and what that means for epsilon values are reversed.

5. After many years of debate with this community using all our datasets in common and often coming to opposite conclusions, I still do not find the evidence that others cite – and including the analysis of the new data here in Section 3.4 ("Share of photolysis and photooxidation) – that the N2O isotopic compositions suggest that O(1D) accounts for a significantly larger share of N2O destruction at low altitudes.

(a) As the authors themselves point out, the 'normalization' and/or 'age correction' will have the largest noise and uncertainty in the lower stratosphere. Much of the variations in Figure 10 could be noise.

(b) I do not find Figure S3b to be convincing or perhaps even relevant. It is not the 'share' of N2O destruction by photolysis versus reaction with O(1D) that is relevant (which is what is plotted in Fig S3b), if the lifetime of N2O with respect to destruction to both processes is 70 years. 70 years is the 'in situ' N2O lifetime at 18 km in the tropics, for example. Thus, the decrease in N2O mixing ratios observed just above the tropical

tropopause is due to photochemical destruction elsewhere: photochemically aged air from middle latitudes (which itself came from higher altitudes) is transported into the lower tropical stratosphere. . .. In other words, if essentially no N2O destruction is occurring in the lower stratosphere, then what does it matter if the share of photooxidation by O(1D) is >70% according to the ratio of their rate constants/J-values (as in Figure S3b)? Likewise, what does it mean in Figure S3b in February above the Arctic Circle when and where it is completely dark to show the relative rates of destruction of N2O by photolysis and by O(1D)? And, finally, the tropical tropopause is at 18+ km in the deep tropics, so the share of O(1D) photoxidation below 18 km doesn't really matter, does it, if this tropospheric air keeps being 'reset' by tropospheric N2O? Hence, I don't see the relevance of Fig S3b, unless it is also accompanied with a plot of the total rate of destruction of N2O as a function of latitude and altitude to be able to better interpret Fig S3b and the point the authors are trying to make.

(c) Another reason why the datapoints may shift nominally toward the 'photooxidation' fractionation line in Figure 10 at the lowest altitudes is that some of these samples were likely collected in 'lowermost stratosphere' (potential temperature < 380K) where there is substantial mixing with tropospheric air that has not entered the stratospheric overworld across the tropical tropopause (or theta>380K). The epsilon apparent could be even LOWER, without resorting to invoking oxidation by O(1D) – if it's just mixing with tropospheric air. In this case, it might not be true that the ratios of epsilon apparent values are really independent of transport. I am not claiming that all the deviations are from the lowermost stratosphere but I would just caution that there are several reasons (a, b, and c) that I would be very careful about and try to rule out, if possible, when interpreting the data points and their spread in Figure 10.

(d) What I would find most convincing (in addition to being very careful about the possibilities in a-c) is the following: Use the 3D model and try two things. 1. Set the O(1D) KIEs in the model to 1 and see if the model shows anything different for the ratio of epsilon apparents for different N2O isotopes vs N2O mixing ratio relationships (best)

or the ratio of epsilon apparents for different N2O isotope versus altitude relationships (ok) without any isotope fractionation for N2O+O(1D) versus their normal isotope runs. If the results for the two different types of runs look the same, then O(1D) isotope fractionation could not have been contributing anything in the lower stratosphere. Indeed, Morgan et al. [2004] did just this and didn't find any evidence in the modeled N2O isotopes for an increased fraction of N2O destruction by reaction with O(1D). I think this is significant. 2. Another thing to try is what Park et al. [2004] suggested; they agreed with McLinden [2003] and a number of non-isotope N2O studies and models that suggest there is not a larger role for N2O destruction in the lower stratosphere beyond about 10% of the total destruction. However, because the KIEs for O(1D)+N2O are quite small, perhaps an increased contribution could be in the noise also in comparing the model results for the two different scenarios with and without O(1D) KIEs. Park et al suggested that another complementary model test of this idea would be to compare the ratios of epsilon values in regular model results versus model results in which O(1D) photooxidation had been removed all together and to examine any differences in the lower stratosphere, in combination with model results with the O(1D) KIEs set to 1.

6. Finally, as noted above, the authors could provide some more background and motivation for why tropical data is exciting and interesting as well as any practical applications throughout. Some more specific suggestions:

* It would be beneficial for reproducibility and contribution to the scientific community if the authors included a table in the supplementary information of their data and a table of derived epsilon values and uncertainties.

* Isotope Ratio Mass Spectrometry (there's not really any 'Monitoring' in this, yet 'monitoring appears in several places in this acronym).

* In Methods, a few more details could be more explicit: -> there is only one or later two samples per small balloon flight

-> the fact that a sample covers an altitude range rather than one altitude needs to be included explicitly in the methods section, and what a typical altitude range is (and if it depends on altitude).

_> The Griffith et al FTIR is technically a 'remote' measurement, not 'in situ'. They are retrieving information rather far from the gondola, even though the gondola is flying on a huge balloon in the stratosphere.

* The 'age corrections': beware the seasonal cycle of $CO_2$ that propagates from the troposphere. This can give a large error in the estimate mean age [e.g., Andrews et al., 1999, 2001].

* Air enters the stratosphere from the tropics, and, based on $CO_2$ mixing ratio measurements taken over a very long time period (years) in both hemispheres, the air has a composition reflecting an admixture of NH and SH air [e.g, Boering et al. 1996, Andrews et al. 1999, 2001]. To take Cape Grim as the 'boundary condition' for SH stratosphere measurements and Mace Head for NH stratosphere measurements and a tropical site for tropical stratosphere measurements seems unduly complicated (if this correction is small) or simply introduces more error (if this 'correction' is actually significant). As noted above, also, it would be nice to have the raw data in supplementary materials as well as the 'corrected' and 'normalized' data used in the analyses.

* Section 2.3 model details: This section is difficult to read and to follow what is the same from earlier publications and what is different.

* Page 5, Lines 28-30: Why aren't lab data used for 18O epsilon values? This passages makes it sound like they are derived from stratospheric observations, or am I misunderstanding something? "Fractionation of $N_2O$ isotopocules was simulated using wavelength-dependent and temperature-dependent enrichment factors for 14N15N16O and 15N14N16O of von Hessberg et al. (2004) and that of 14N14N18O estimated from the relation 30 between apparent epsilon for each isotopocule observed in the stratosphere."

* Page 7, Line 1: why are the other isotopes not plotted? Could they be put in supplementary figures?

Page 7, Line 7: It should be clarified that these three data points are exactly what would be expected for tropical high altitude data – epsilon effective $\sim$ epsilon sink and why. . ..

* Page 7, Lines 15-20: This has been noted in many studies before. Here and in other places, the manuscript could be more complete in referencing previous studies, their conclusions, and similarities and differences with this study.

* Page 7, Line 19: Could the values in Fig 4 (and the additional breakdown of the Esrange Sweden flight to "M" and "L") be given as a table (with error bars) in supplementary materials? I'd like to see the numbers, for comparison with error bars and for comparison with future and past studies (e.g., with past aircraft samples in Park et al. [2004] and Kaiser et al [2006] that are not included in this manuscript). Also, why do the 17N data (HBD) not appear in the figure? [In addition, the gold color for photolysis was a bit confusing, because gold was used in the previous figures for the subtropical 17N data!]

* Section 3.3: (in addition to major comments above): More discussion (even if very brief) of previous studies and their conclusions that agree with, or do not, with this study is needed.

* Page 8, Line 1-4: "There is no bending or curved structure apparent in the Rayleigh diagram, which suggests that chemical processes are not variable, although the small fluctuation in the lower left region in Fig. 5a will be discussed later." First, "chemical processes are not variable" will not be clear to many readers as to what the authors are trying to communicate regarding the variability of the processes involved. Second, in addition to my not agreeing that the lower left region of Fig 5a means something about O(1D) chemistry dominating photolysis (see above major comments), I also find this statement misleading: Chemical processes can vary throughout the stratosphere but the isotopic composition of N2O – being a long-lived tracer – is not necessarily

going to retain a 'memory' of this since air with many different transport histories (and chemical histories) are mixed together; in fact, N2O destruction is dominated by photolysis in upper stratosphere, and then the remaining N2O is redistributed from the upper stratospheric sink region. So I don't think the authors can make the inference the other way around – that if there is no bending or curvature in the Rayleigh plot, that that means that chemistry and relative chemical time scales do not vary throughout the stratosphere.

* If the authors do wind up using their data to support a large relative role for photooxidation in the lower stratosphere, then they need to more explicitly differentiate their comments on Page 9, line 25 that "transport and mixing" are discussed as the difference between the epsilon values at different the latitudes, from those on Page 10, Line 4 that photooxidation is cited as having a major influence in the lower stratosphere (for which it is the ratios of epsilon values are assumed to be independent of transport of mixing); this is confusing without clarification (for which background is lacking in general for this "ratio of ratio" approach).

* Throughout: "Transport" is the term used to describe air mass or tracer transport (while transportation is buses, cars, airplanes, etc.)

* By the way, the so-called "transport barrier" is not really a barrier – it just looks like one in satellite views of tracers like the aerosol from the Mt Pinatubo eruption! It 'looks' and 'acts' like a barrier because, between ∼22 and 28 km, the vertical ascent rates in the tropics are much faster than quasi-horizontal transport rates out of the tropics to the midlatitudes, which in turn are much much faster then quasi-horizontal rates from the midlatitudes into the tropics (and contrasting with the rate of transport and mixing into the tropics below 22 km, which your data here, and many other types of tracer studies, are completely consistent with!). Alan Plumb discusses this quite a bit, as 'barrier' is a bit of a misnomer when one is contemplating the dynamics and transport.

[Figure]

2017.

---

## Author Comment (AC1) · 21 Aug 2017

Reply to the referee #1

Referee's comment is typed in blue, and authors' response is typed in black.

This paper extends previous measurements and analysis of the vertical distribution of N2O isotopocules into the tropical stratosphere. It introduces a new sampling system with a much weight-reduced cryogenic sampler. The measurements are impressively accurate and precise and show systematic differences to previously published mid latitude profiles, with implications for the balance between photolysis and photooxidation of N2O, and transport, in determining the vertical distributions. The paper is clearly written and presented. It is well suited to ACP and I recommend publication as is except for a few technical edits as listed below.

We appreciate the referee for his/her constructive comments. We have revised the manuscript according to his/her suggestions.

Technical comments:

P3 L11: …but it has not been FULLY examined because of…

Revised as suggested. (P3 L22 in the revised text)

P8 L13 and elsewhere – please replace all instances of "transportation" with "transport".

Revised as suggested.

P8 L14 & 15: I suggest using "photochemistry" rather than "photolysis" here because the first reference on L14 is to both photolysis and phot-oxidation.

This paragraph has been deleted according to the comments by referee #2.

P8 L19: "faster" would be more unambiguous than "larger" in this context

This paragraph has been deleted according to the comments by referee #2.

Caption figure 2 – it would be helpful here to list the site names and three letter abbreviations so the reader is not obliged to go to the supplementary material to find out. Defined once is enough to cover all later figures.

Revised as suggested.

---

## Author Comment (AC2) · 21 Aug 2017

Reply to the referee #2

Referee's comment is typed in blue, authors' response is typed in black, and the change in the revised text is highlighted with red.

In this manuscript, Toyoda et al present the first vertical profiles of the isotopic composition of N2O in the deep tropical stratosphere (i.e., quite literally over the equator) that extend above 21 km. They use a new, unique very small cryogenic whole air sampler that can be launched from ships using small balloons and retrieved from the ocean surface after a water landing; this is critical since more traditional cryogenic whole air samplers are extremely heavy and therefore must be launched with very large balloons, greatly limiting their deployment in the deep tropics, especially if they must also be designed to survive ocean landings in addition to the difficulty of finding suitable launch sites near the equator. The magnitudes of the isotopic fractionation of N2O in the deep tropics and the character of the relationships between the N2O isotopic compositions with N2O mixing ratios have been a topic of speculation for over a decade, especially given the promise that deep tropical data up to 30 to 35 km could hold for further elucidation of the patterns and rates of transport between the tropics and extratropics and for the competition between transport and photochemistry in determining the isotopic composition of N2O. This first new deep tropical dataset with measurements up to 30 km is therefore exciting and interesting. These data, their comparison with previous measurements and model results from a number of research groups, and their analysis will therefore be of great interest to the stratospheric and isotope communities of researchers. That said, there are a number of improvements to the analysis, discussion, and presentation of the data that I think should be considered before publication.

We appreciate the referee for his/her many critical and constructive comments. We have revised the manuscript according to his/her suggestions.

Major points to consider:

1. A more clear motivation for the need for and applications of N2O isotope observations in the tropics could be given in the introduction as well as in the discussion and conclusions. The context and questions that remain unanswered or are controversial (or at least that some of us think are controversial and not proven) are not clearly outlined or discussed in the manuscript as currently laid out. In addition, many other tracer measurements have been used to study transport and mixing – what might be unique about N2O isotopocules? This could be mentioned, especially in the concluding section.

In the introduction, we revised or added following sentences describing the motivation, context, remaining questions, and uniqueness of $N_2O$ isotopocules.

P2, L18–20

Mixing ratios of trace gases such as $N_2O$ and their interrelationships are regarded as useful tools to establish a detailed picture of stratospheric circulation (Plumb, 2007), and previous observations showed compact tracer relationships that depend on latitude (e.g, Michelsen et al., 1998).

P2, L25–28

In the context of stratospheric distribution, isotopocule ratios are unique in their ability to provide the degree of photochemical decomposition and the relative importance of the above-mentioned two decomposition pathways (Toyoda et al., 2001; Röckmann et al., 2001).

P3, L23–27

As noted above, tropical stratosphere is the starting point of meridional transportation of $N_2O$ injected from the troposphere and the rates of photochemical reactions are faster than those in the extratropics because of stronger actinic flux. Although there are a few reports on vertical profiles of $N_2O$ isotopocules over India (18°N) (Röckmann et al., 2001; Kaiser et al., 2006), isotopic composition of $N_2O$ in upwelling tropical air has not been characterized.

P3, L29–P4, L4

Another controversial problem about the stratospheric $N_2O$ is whether the photooxidation sink (Eq. 2) has larger contribution than 10% in the lower stratosphere. Kaiser and coworkers found that the ratios of enrichment factors of isotopocule during photolysis and photooxidation are distinct (Kiaser et al., 2002a) and they estimated that a much larger fraction than 10% is removed by photooxidation at least in the lower stratosphere ($N_2O$ mixing raitos > 300 nmol $mol^{-1}$) (Kaiser et al., 2006). However, similar but a little simplified analyses by Park et al. (2004) and Toyoda et al. (2004) could not detect significant differences in the enrichment factor ratios between the lower and middle stratospheres, and Park et al. (2004) speculated that the enrichment factor ratios could be affected not only by the relative share of the two sink pathways but also by other factors such as transport.

In the concluding section, we revised the sentences to clarify the points revealed by our new observations as follows.

P11, L24–P12, L5

Vertical profiles of isotopocule ratios of $N_2O$ in the equatorial stratosphere are found using balloon-borne compact cryogenic samplers and mass spectrometry in the laboratory. This report of the relevant literature is the first describing observations of them over the equator. Unlike other region of the stratosphere, enrichment factors for isotopocules in the middle equatorial stratosphere (25–30 km, or $[N_2O]$ < 260 nmol $mol^{-1}$) agreed with those obtained with laboratory photolysis experiments, suggesting that the isotopocule ratios are determined mainly by photolysis because of weak vertical or horizontal mixing in the tropical upwelling. In the lower equatorial stratosphere (< 25 km or $[N_2O]$ > 260 nmol $mol^{-1}$), isotopocule ratios suggest that differently aged air masses are mixed because of the meridional transport and that decomposition by photooxidation might also plays a significant role. Vertical and latitudinal distributions of $N_2O$ and its isotopocules are found to be a unique tool to diagnose the relationship between photochemistry and transport in the stratosphere. Further

observations of temporal variations and comparison with ACTM simulation will be needed to examine the change in the meridional circulation and obtain the quantitative estimate of the importance of photooxidation pathway.

2. A lot of detail is given to 'normalize' the N2O isotope and mixing ratio data since there is quite a large time span of datasets with which the new tropical observations are compared. There are also ratios of epsilon values, which in some cases may be referred to as 'normalized' (i.e., relative to the tropopause). It is unclear to me what 'age corrected' and 'normalized' mean in various places. And it is also unclear which data are plotted in the main text and in the supplementary material. All this information should be made more explicit and clear in both the main text and the figure captions. I think it is also important to plot up and/or give in tables both the 'raw' data as well as the age-corrected/normalized data for comparison. I have further specific comments on this topic below.

The 'normalized' means that mixing ratio or delta values are expressed as relative values against those at the time when the measured air mass existed in the troposphere. We note that neither 'age corrected' nor "normalized epsilon" appear in our manuscript. The main text and the figure captions have been revised as follows, avoiding the ambiguous expression like "normalized $\delta$". Also we added Table S3 to present raw data as well as converted data.

P8, L2–3

In Fig. 3, the $\delta^{15}N^{bulk}$ is shown against the $N_2O$ mixing ratio after the normalization described in Eq. 6 (Rayleigh plot).

P9, L5–9

The ratio of $\varepsilon$ values for independent isotopocules (e.g., $\varepsilon(^{15}N^{bulk})/\varepsilon(^{18}O)$), however, has been identified as a useful parameter to distinguish photolysis and photooxidation (Kaiser et al., 2002a) because its sensitivity to wavelength and temperature is small and it is not affected by mixing process. Figure 5 shows the data obtained in this study and some previous ones in $\delta$–$\delta$ space after the normalization described in Eq. 6.

P11, L2–10

Kaiser et al. (2006) used the ratio of $\varepsilon$ values for $^{15}N^{bulk}$ and $^{18}O$ ($\psi$) and the ratio of $\varepsilon$ values for $^{15}N^{\alpha}$ and $^{15}N^{\beta}$ ($\eta$) to estimate the relative share of photolysis and photooxidation based on the fact that $\psi$ and $\eta$ are almost independent of transport processes and are significantly different between the two decomposition processes. They computed $\psi$ and $\eta$ values directly for each individual sample in order to avoid statistical errors associated with linear regression to the $\delta$–$\delta$ plot which was adopted by Toyoda et al. (2004) and Park et al. (2004). In Fig. 10, we show $\psi$ and $\eta$ values calculated using the data presented in Fig. 2 in the manner similar to that of Kaiser et al. (2006) except that we used

individual date of stratospheric entry for each data to normalize the δ values instead of using a single tropopause date. Although it is noteworthy that errors in $\psi$ and $\eta$ values increase concomitantly with decreasing altitude because of the decrease in the $\delta$ values, ….

Caption of Figure 3

… Both parameters are normalized to their values at the time when the corresponding air mass entered the stratosphere (see Eq. 6 in the text).

Caption of Figure 10

Vertical profiles of ratio of $\varepsilon$ values for $^{15}N^{bulk}$ and $^{18}O$ ($\psi$) and the ratio of $\varepsilon$ values for $^{15}N^{\alpha}$ and $^{15}N^{\beta}$ ($\eta$) calculated in the manner similar to that of Kaiser et al. (2006).

3. My own opinion is that analyzing the isotope data in subsets of altitude bins (which depend on latitude) is a lot less illuminating than with respect to grouping by N2O mixing ratio, which is in some ways less complicated than altitude given the Brewer-Dobson circulation bringing much older air down from higher altitudes in the extratropics and the polar vortices. Indeed, by binning by N2O mixing ratio, aircraft data from other researchers and I'm sure from the authors' own aircraft datasets can be more easily compared, rather than being limited here by vertical profiles obtained by balloons. This might also be why the Esrange flight from Sweden in early February was not split into a 'lower' and a 'middle' because figuring out where to put the altitude bin might have been difficult for this flight since N2O is so low at a given altitude relative to the other balloon profiles. For example, the data plotted in Figure 3 for the Swedish vortex flight beg to be plotted into 2 groups "M" and "L" (for 'middle' and 'lower' respectively), and yet they are given as one group. It would be interesting to see how small the apparent epsilon value would be for the vortex data, as well as to compare that with remnant vortex data on samples collected by aircraft from Park et al. [2004]. The more complicated the transport and mixing history of air masses in different regions of the stratosphere and/or at different times, the lower the apparent epsilon value is, and the Esrange data and some aircraft data could be very convincing in showing this, and contrasting it with these first deep tropical data at altitudes >23 km and N2O mixing ratios <250 ppb, for which transport and mixing have the least influence compared with the rest of the stratosphere. This is the point the authors are trying to make but I think it could be made much more clear.

We thank the referee for the constructive comment. We analyzed the data in subset of altitude bins simply because all of our data were obtained as vertical profiles. We agree that it would be better to use $N_2O$ mixing ratio for the grouping to compare with data obtained by aircrafts. Even so, we consider the normalization is necessary to account for the increasing trend of tropospheric $N_2O$ and the time lag from the entrance into the stratosphere, instead of simply using mixing ratio. In the revised manuscript, we use $-\ln\{[N_2O]/[N_2O]_0\} = 0.2$ and 0.6 as a boundary for tropics and extratropics, respectively, because at these points slope of Rayleigh plots change significantly and the highest $R^2$ values are obtained in most cases. Based on the new binning, ε values in the middle and lower stratosphere of the winter arctic polar vortex

are now also shown. Abstract, a few sentences in section 3.2, conclusion and Figure 4 have been revised accordingly.

4. The exciting thing about the new deep tropical dataset is that it clearly shows what we all expected: that in the deep tropics where the stratosphere is effectively the most "isolated" relative to all other regions, the fractionation for all the N2O isotopocules is as "Rayleigh" as it gets, and that the apparent epsilon values are close to that for photolysis in the lab. However, rather than discuss this qualitatively from a "Rayleigh" perspective, and using other stratospheric data/observations to show that this is consistent with what we know about relative transport and chemical time scales in different regions (and seasons) in the stratosphere, the authors instead jump right to an equation from Kaiser et al [2006] (which also appears in Kaiser et al. 2002a, an observation and lab study, and Morgan et al. 2004, a modeling study) that is an approximation to an approximation for a 1D model of the stratosphere. This would, I suppose, be all right except that – given what we know about transport time scales and the photochemical lifetime of N2O in the tropics and the extratropics as a function of altitude from models and from other stratospheric observations (such as N2O, CO2, H2O, CH4 and other long-lived tracers and their correlations) – this 1D model analogy I think actually gives us the wrong answer! Thus, I find this section misleading and not especially useful or illustrative at best and incorrect at worst.

Here is my point as to why I think simple departures from a Rayleigh model based on how much transport and mixing there has been relative to other regions of the stratosphere are more enlightening than this pseudo 1D model treatment: First, below 22 km, transport time scales are fast (and the pseudo 1D treatment convolves both vertical and horizontal/quasi-isentropic transport as 'eddy diffusion') and the photochemical decomposition of N2O is very slow (by both photolysis and reaction with O(1D)). At 20 km, the transport time scale may be months while the e-folding time for N2O destruction is 70 years, for example. If you plug t(trans)«t(chem) into Equation 7, however, the result is epsilon effective = epsilon sink. In contrast, higher up in the tropical stratosphere, t(trans)»t(chem) [where the photochemical destruction really starts picking up speed], so now Equation 7 simplifies to epsilon effective = 1/2 epsilon sink. But that's exactly where one would expect Rayleigh fractionation to hold – where the change in N2O isotopic composition is going to follow the change in N2O mixing ratio according to epsilon for the sink without transport messing things up. So these two limits from Equation 7 are the opposite of what the new deep tropical observations show! At midlatitudes below 30 km where t(trans)«t(chem) for N2O [again, from many other stratospheric 'non-isotope' studies], Equation 7 would give epsilon effective = epsilon sink and yet the observations show it is more like the other limit, epsilon effective = 1/2 epsilon sink. So also the opposite of the Equation 7 limits. Indeed, even Kaiser et al. [2006] say the following: "even though the apparent fraction constants could possibly be reconciled with certain diffusion coefficients and [N2O] destruction rates, the [observed] increase of the magnitude of the apparent fractionation constant with altitude is actually the opposite to what one would expect from a simple vertical reactiondiffiusion-advection model. . . Despite the conceptual usefulness of this 1D description, it has to be

stressed that stratospheric transport cannot be characterized as a function of altitude alone and that vertical diffusion is actually not an important process, even though it can be helpful to describe stratospheric transport to some extent. Meridional transport schemes have to be included to explain the variation of epsilon apparent." Thus, Equation 7 gives the opposite answer to all the N2O isotope observations I am aware of, including these new ones in the deep tropics, and even Kaiser et al [2006] had to invoke what we know about meridional patterns of transport and mixing to explain all the stratospheric observations available in 2006. Thus, it is not true, as stated by the authors in this manuscript on Lines 15-20 on Page 8 in this section, that Kaiser et al "used this simplified scheme in Equation to describe qualitatively the vertical and meridional trends in epsilon [values] obtained over latitudes ranging from 18 to 80 (N or S)." Hence I do not find Equation 7 nor the discussion around it to be useful and I think it is actually wrong to use it here. The 2D nature of the Brewer-Dobson circulation and the concomitant chemical and transport timescales associated with the transport and mixing in 2D is required to understand the distribution of N2O isotopes and their relationship with N2O mixing ratios everywhere; Equation 7 does not work.

That said, the 2D mixing scheme the authors use does qualitatively work for me. While I'm not sure where their '10%' number comes from (it's arbitrary), qualitatively it can explain the 2D mixing patterns relative to the photochemical time scales for N2O destruction and why (a) at higher altitudes in the tropics, epsilon apparent = epsilon sink (again a very exciting set of observations) and (b) at lower altitudes in both the tropics and the extratropics, epsilon apparent is more like 1/2 epsilon sink). Thus, I recommend simply removing the 1D model equation and discussion entirely. It leads the reader to the opposite conclusion, as has been pointed out previously, while the qualitative mix-ing model is actually more realistic (or at least illustrative), despite it perhaps seeming somewhat arbitrary to some readers.

And the 3D model comparison is a nice follow-on to the 2D results and discussion, and shows quite a nice quantitative agreement with the observations in Figures 8 and 9. In fact, the authors conclude that "Because in situ epsilon photolysis used in the model calculation is nearly the same between low and high latitudes (Fig. S3a), this agreement supports the inference that the major causes of the difference [in epsilon values] are transport and mixing." I agree, and Morgan et al. [2004] have similar results from their 2D CTM results.

The 1D Equation 7 conclusions being the opposite of what these and all past data (I think) show, if the authors modify their approach on this, then these ideas also need to modified in the abstract and conclusions as well, where in some cases the opposite of 'transport' vs 'chemical time scales' and what that means for epsilon values are reversed.

We thank the referee for the critical comments and detailed explanation. We realize that the 1D model is inappropriate to explain the difference in ε values between the lower and middle stratosphere. We have revised section 3.3.2 as follows, removing the discussion with 1D model.

P9, L15–24

Transport processes accompanied by mixing of variously aged stratospheric air has been considered as the major cause of lower $|\varepsilon|$ value in the stratosphere than in the laboratory photochemical decomposition (Park et al., 2004; Kaiser et al., 2006). Our new observation revealed that all the $N_2O$ isotopocules are fractionated by the almost ideal Rayleigh process in the middle stratosphere over the deep tropics where the stratosphere is effectively the most isolated relative to all other regions. This underlines how much transport and mixing affect the apparent $\varepsilon$ value.

We then consider the effect of transport on the apparent $\varepsilon$ at different latitudes with a conceptual two-dimensional circulation model in the tropical and extra-tropical stratosphere that was proposed to explain tracer–tracer correlation (Plumb, 2002). …

5. After many years of debate with this community using all our datasets in common and often coming to opposite conclusions, I still do not find the evidence that others cite ‐ and including the analysis of the new data here in Section 3.4 ("Share of photolysis and photooxidation) – that the N2O isotopic compositions suggest that O(1D) accounts for a significantly larger share of N2O destruction at low altitudes.

(a) As the authors themselves point out, the 'normalization' and/or 'age correction' will have the largest noise and uncertainty in the lower stratosphere. Much of the variations in Figure 10 could be noise.

The larger errors in $\psi$ and $\eta$ at lower altitudes are not introduced by the normalization, but the small difference in $\delta$ values between the stratosphere and troposphere. Effect of the normalization is smaller than the error bars in Fig. 10 as shown below.

[Figure]

(b) I do not find Figure S3b to be convincing or perhaps even relevant. It is not the 'share' of N2O destruction by photolysis versus reaction with O(1D) that is relevant (which is what is plotted in Fig S3b), if the lifetime of N2O with respect to destruction to both processes is 70 years. 70 years is the 'in situ' N2O lifetime at 18 km in the tropics, for example. Thus, the decrease in N2O mixing ratios observed just above the tropical tropopause is due to photochemical destruction elsewhere: photochemically aged air from middle latitudes (which itself came from higher altitudes) is transported into thelower tropical stratosphere. . .. In other words, if essentially no N2O destruction is occurring in the lower stratosphere, then what does it matter if the share of photooxidation by O(1D) is >70% according to the ratio of their rate constants/J-values (as in Figure S3b)? Likewise, what does it mean in Figure S3b in February above the Arctic Circle when and where it is completely dark to show the relative rates of destruction of N2O by photolysis and by O(1D)? And, finally, the tropical tropopause is at 18+ km in the deep tropics, so the share of O(1D) photoxidation below 18 km doesn't really matter, does it, if this tropospheric air keeps being 'reset' by tropospheric N2O? Hence, I don't see the relevance of Fig S3b, unless it is also accompanied with a plot of the total rate of destruction of N2O as a function of latitude and altitude to be able to better interpret Fig S3b and the point the authors are trying to make.

We thank the referee for this critical comment. We have added a supplementary graph of total loss rate constant of $N_2O$ as a function of altitude (Fig. S6a, note that figure number has been changed from S3) and deleted a part of the plots of tropics and all the plots of February Arctic region so that the plots show the parameters in the region where photochemical $N_2O$ loss really occurs. We agree that the loss rate of $N_2O$ in the lower stratosphere is very slow and therefore the larger share of $O(^1D)$ sink in *in situ* total loss below 25 km is not directly reflected on the observed isotopocule ratios. However, we consider the decrease in $\psi$ and $\eta$ in the lower stratosphere cannot be explained solely by transport and mixing based on an example using a conceptual 2D model (see our response to the comment #5(c) below). As the referee pointed out, majority of $N_2O$ injected into the stratosphere is destructed in the middle stratosphere and the photochemically aged air mass must be transported into the lower stratosphere. But we consider there is a possibility of additional decomposition of remaining $N_2O$ during the transport (which should be slower than that of the tropical upwelling) and the isotopic signature of $O(^1D)$ pathway could be imprinted. Sentences in the last half of section 3.4 have been revised as follows.

P11, L13–22

Although the loss rate of $N_2O$ in the lower stratosphere is very slow and the majority of $N_2O$ injected into the stratosphere is photolyzed in the middle stratosphere as noted by Park et al. (2004), the share of photooxidation in *in situ* total loss increases in the lower stratosphere (Fig. S6b). Therefore, there is a possibility of additional decomposition of remaining $N_2O$ during the transport (which should be slower than that of the tropical upwelling) and the isotopic signature of $O(^1D)$ pathway could be imprinted, and the photochemically aged air mass must be transported into the lower stratosphere of the tropics and extratropics. However, Morgan et al. (2003) reported that inclusion of isotope fractionation for photooxidation into their 2D model does not make a significant contribution to overall fractionation in the stratosphere, and Park et al. (2004) discussed an alternative modelling

approad with and without O($^1$D) sink to test the importance of O($^1$D) reaction. Further studies using 3D model would be necessary to solve this controversial problem.

(c) Another reason why the datapoints may shift nominally toward the 'photooxidation' fractionation line in Figure 10 at the lowest altitudes is that some of these samples were likely collected in 'lowermost stratosphere' (potential temperature < 380K) where there is substantial mixing with tropospheric air that has not entered the stratospheric overworld across the tropical tropopause (or theta>380K). The epsilon apparent could be even LOWER, without resorting to invoking oxidation by O(1D) – if it's just mixing with tropospheric air. In this case, it might not be true that the ratios of epsilon apparent values are really independent of transport. I am not claiming that all the deviations are from the lowermost stratosphere but I would just caution that there are several reasons (a, b, and c) that I would be very careful about and try to rule out, if possible, when interpreting the data points and their spread in Figure 10.

We agree that apparent epsilon values in the lower stratosphere could be decreased by mixing with tropospheric air, but we cannot agree that the ratios of apparent epsilon values might depend on transport. For example, if we make the 2D simulation shown in Fig. 7 with different epsilon value (e.g., −40 permil) and calculate the ratio of apparent epsilons obtained for the black stars by the two different simulations, we obtain exactly the same ratios for all data points. As the referee points out below, 3D simulation might be needed to further ascertain this, but we consider it is beyond the scope of this paper.

(d) What I would find most convincing (in addition to being very careful about the possibilities in a-c) is the following: Use the 3D model and try two things. 1. Set the O(1D) KIEs in the model to 1 and see if the model shows anything different for the ratio of epsilon apparents for different N2O isotopes vs N2O mixing ratio relationships (best) or the ratio of epsilon apparents for different N2O isotope versus altitude relationships (ok) without any isotope fractionation for N2O+O(1D) versus their normal isotope runs. If the results for the two different types of runs look the same, then O(1D) isotope fractionation could not have been contributing anything in the lower stratosphere. Indeed, Morgan et al. [2004] did just this and didn't find any evidence in the modeled N2O isotopes for an increased fraction of N2O destruction by reaction with O(1D). I think this is significant. 2. Another thing to try is what Park et al. [2004] suggested; they agreed with McLinden [2003] and a number of non-isotope N2O studies and models that suggest there is not a larger role for N2O destruction in the lower stratosphere beyond about 10% of the total destruction. However, because the KIEs for O(1D)+N2O are quite small, perhaps an increased contribution could be in the noise also in comparing the model results for the two different scenarios with and without O(1D) KIEs. Park et al suggested that another complementary model test of this idea would be to compare the ratios of epsilon values in regular model results versus model results in which O(1D) photooxidation had been removed all together and to examine any differences in the lower stratosphere, in combination with model results with the O(1D) KIEs set to 1.

We appreciate the referee for the constructive comment. Before considering his/her suggestion, we have

examined how the $\psi$ and $\eta$ values calculated from the data simulated by the 3D model are compared with those from observation. As shown below, $\eta_{model}$ shows similar vertical trend as $\eta_{obs}$ although its range is smaller. In contrast, $\psi_{model}$ is almost independent of the altitude and shows larger value in the tropical lower stratosphere. We also confirmed that $\psi$ and $\eta$ calculated from $\varepsilon$ values based on in situ J values do not depend on latitude nor altitude. Therefore, we recognize that there is a possibility that $\psi$ and $\eta$ might be affected by factors other than the relative share of photolysis and photooxidation.

We then examined the procedure how to conduct the simulations suggested by the referee. Unlike the simpler 2D model of Morgan et al. (2004), our 3D model optimizes emissions from the individual surface $N_2O$ sources using several emission scenarios to simulate the tropospheric $N_2O$ mixing ratio and isotopocule ratios. Therefore, if we change epsilon for photooxidation to 0‰ (i.e., KIE = 1) or if we remove the photooxidation sink, tropospheric (and hence tropical lower stratospheric) mixing ratios and isotopoule ratios and their trends will be also changed. To keep the tropospheric values close to the realistic ones, additional optimization of the surface emission are needed and this will cost huge computational resources (output data size will be 3.4TB for each simulation) and time (more than three weeks for each simulation). We consider that this is too much task to revise our current manuscript and that it should be done as a separate work.

[Figure]

6. Finally, as noted above, the authors could provide some more background and motivation for why tropical data is exciting and interesting as well as any practical applications throughout. Some more specific suggestions:

6-1) It would be beneficial for reproducibility and contribution to the scientific community if the authors

included a table in the supplementary information of their data and a table of derived epsilon values and uncertainties.

We have added two supplementary tables. Raw and normalized isotopic data are listed in Table S3. Derived epsilon values are listed in Table S4.

6-2) Isotope Ratio Mass Spectrometry (there's not really any 'Monitoring' in this, yet 'monitoring appears in several places in this acronym).

We have revised the sentence as follows.

P5, L10–12

The isotopocule ratios, … using gas chromatography – isotope ratio mass spectrometry.

6-3) In Methods, a few more details could be more explicit: -> there is only one or later two samples per small balloon flight

We have revised the sentences as follows.

P4, L25–27

For each balloon flight, a 5–8 L STP of air sample was collected by a single sampler at programmed altitude of 19–29 km.

P4, L30–31

For each balloon flight, two samplers integrated into a single gondola were launched from …

6-4) -> the fact that a sample covers an altitude range rather than one altitude needs to be included explicitly in the methods section, and what a typical altitude range is (and if it depends on altitude).

We have added the following sentences.

P5, L4–6

Sampling was conducted while the balloon was ascending except the flight on Feb 5, 2012. Typical altitude range was about 2 km, and we took the central value of the range as the sampling altitude.

6-5) _> The Griffith et al FTIR is technically a 'remote' measurement, not 'in situ'. They are retrieving information rather far from the gondola, even though the gondola is flying on a huge balloon in the stratosphere.

We have revised the sentence as follows.

P3, L1–5

For balloon observations, … (b) remote measurements are conducted ….

6-6) The 'age corrections': beware the seasonal cycle of CO2 that propagates from the troposphere. This can give a large error in the estimate mean age [e.g., Andrews et al., 1999, 2001].

According to Andrews et al. (2001), errors for the mean age associated with the tropospheric $CO_2$ seasonal cycle is less than 0.3–0.5 years. This is translated to error of 0.2–0.4 nmol mol$^{-1}$ in the correction for $N_2O$

mixing ratio at the time of the air mass's entry into the stratosphere because annual trend of tropospheric $N_2O$ is ~0.7 nmol mol$^{-1}$/yr. This error is about 25–50% and 5–10% of the correction value in the lower stratosphere (~1.5 nmol mol$^{-1}$ ) and in the middle stratosphere (1.5–4 nmol mol$^{-1}$), respectively. But in the final logarithmic form of ln{$[N_2O]/[N_2O]_0$}, it is less than 0.001. Therefore, the effect of seasonal cycle of $CO_2$ is insignificant in the analysis of present study.

We added the following sentences at the end of section 2.2 to explain the effect of the normalization and possible errors introduced by the data conversion.

P6, L11–16

In Table S3 we compare how much this correction regarding the age of air changed the position of each data point in Figs. 3 and 5. Typically, the term related to mixing ratio ($-\ln\{[N_2O]/[N_2O]_{trp}\}$) is decreased by 0.2–3% when we use $[N_2O]_0$, the value when the air mass actually entered into the stratosphere, instead of the value at the same time of the observation, $[N_2O]_{trp}$. The isotopic terms ($\ln\{(1+\delta)/(1+\delta_{trp})\}$) are either increased or decreased depending on their secular trends, and they are changed by 0.2–3%.

6-7) Air enters the stratosphere from the tropics, and, based on CO2 mixing ratio measurements taken over a very long time period (years) in both hemispheres, the air has a composition reflecting an admixture of NH and SH air [e.g, Boering et al. 1996, Andrews et al. 1999, 2001]. To take Cape Grim as the 'boundary condition' for SH stratosphere measurements and Mace Head for NH stratosphere measurements and a tropical site for tropical stratosphere measurements seems unduly complicated (if this correction is small) or simply introduces more error (if this 'correction' is actually significant). As noted above, also, it would be nice to have the raw data in supplementary materials as well as the 'corrected' and 'normalized' data used in the analyses.

As the referee points out, our selection of tropospheric reference value is somewhat arbitrary and the corrections are comparable or small relative to the measurement error. Nevertheless, we consider it is better approach to make a closer analogy to a Rayleigh fractionation system as noted by Kasier et al. (2006). We added Table S3 to present raw data as well as the converted data. See also our reply to the comment #6-6).

6-8) Section 2.3 model details: This section is difficult to read and to follow what is the same from earlier publications and what is different.

The differences from earlier publications are (1) avoiding the optimization of photolytic enrichment factors, (2) meteorological reanalysis data for nudging, and (3) modified surface emissions. The last paragraph in section 2.3 has been revised as follows.

P7, L4–11

While both the surface emissions and the photolytic isotopocule fractionations were optimized in the earlier work by Ishijima et al. (2015), only the former was optimized in the present study. This is because we considered that it would be better to keep the experimentally determined original isotopocule enrichment factors for the purpose of comparison between the model and the

observations in the stratosphere. Moreover, we found that apparent isotopocule enrichment factors obtained by the model simulation become much closer to those by the balloon observations by replacing the meteorological data from JRA-25 (Onogi et al., 2007) with those from ERA-interim. This is probably because dynamics and chemical reactions in the model was improved by the replacement of the meteorological reanalysis data for nudging. Surface emissions ….

6-9) Page 5, Lines 28-30: Why aren't lab data used for 18O epsilon values? This passages makes it sound like they are derived from stratospheric observations, or am I misunderstanding something? "Fractionation of N2O isotopocules was simulated using wavelength-dependent and temperature-dependent enrichment factors for 14N15N16O and 15N14N16O of von Hessberg et al. (2004) and that of 14N14N18O estimated from the relation 30 between apparent epsilon for each isotopocule observed in the stratosphere."

The referee correctly understands. In our model, epsilon values are incorporated as function of wavelength and temperature. Von Hessberg et al. (2004) presented ε values with wave length resolution of about 1 nm and at two different temperatures, which is suitable for our purpose. However, they did not report on $NN^{18}O$ isotopocule, and we could not find any other reports on wavelength and temperature dependency of $\varepsilon(^{18}O)$ as precise as that of von Hessberg et al. (2004). Therefore, we assumed the constant relationship between $\varepsilon(^{18}O)$, $\varepsilon(^{15}N^\alpha)$, and $\varepsilon(^{15}N^\beta)$ based on our previous observations (Toyoda et al., 2004) and approximated $\varepsilon(^{18}O)$. Detailed procedure to determine the $\varepsilon(^{18}O)$ value is described in Ishijima et al (2015). We slightly modified the following sentence.

P6, L27–31

Fractionation of $N_2O$ isotopocules was simulated using wavelength-dependent and temperature-dependent enrichment factors ($\varepsilon$) for $^{14}N^{15}N^{16}O$ and $^{15}N^{14}N^{16}O$ reported by von Hessberg et al. (2004) although the $\varepsilon$ for $^{14}N^{14}N^{18}O$ was estimated from the relation between apparent $\varepsilon$ for each isotopocule observed in the stratosphere due to the lack of suitable experimental reports.

6-10) Page 7, Line 1: why are the other isotopes not plotted? Could they be put in supplementary figures?
The Rayleigh plots for other isotopocules were omitted because they looked similar to that for $^{15}N^{bulk}$. We have added them as Figures S3, S4, and S5.

6-11) Page 7, Line 7: It should be clarified that these three data points are exactly what would be expected for tropical high altitude data – epsilon effective epsilon sink and why. . ..
Although we consider this was described in the next paragraph, we revised it taking account of the comment #6-12).

6-12) Page 7, Lines 15-20: This has been noted in many studies before. Here and in other places, the manuscript could be more complete in referencing previous studies, their conclusions, and similarities and differences with this study.

We added the following sentences in this paragraph.

P8, L22–25

Although the similar latitudinal and altitudinal dependence of $\varepsilon$ has been reported previously for the latitudes ranging from 18°N to 89°N (Park et al., 2004; Kaiser et al., 2006), our equatorial data showed that the change in $\varepsilon$ occurs at altitude with higher $N_2O$ mixing ratio and the $\varepsilon$ value is exactly what would be expected during the $N_2O$ photolysis.

6-13) Page 7, Line 19: Could the values in Fig 4 (and the additional breakdown of the Esrange Sweden flight to "M" and "L") be given as a table (with error bars) in supplementary materials? I'd like to see the numbers, for comparison with error bars and for comparison with future and past studies (e.g., with past aircraft samples in Park et al. [2004] and Kaiser et al [2006] that are not included in this manuscript). Also, why do the 17N data (HBD) not appear in the figure? [In addition, the gold color for photolysis was a bit confusing, because gold was used in the previous figures for the subtropical 17N data!]

Figure 4 has been revised as we described in response to the comment #3. Also we corrected a mistake in plotting epsilon values for photolysis. The values in the figure are given in newly added Table S4. The HDB data were not included because the highest sampling altitude is lower than other observations and it was difficult to estimate epsilon value for middle stratosphere.

6-14) Section 3.3: (in addition to major comments above): More discussion (even if very brief) of previous studies and their conclusions that agree with, or do not, with this study is needed.

The first paragraph in section 3.3.2 has been revised referring to two previous studies (see our response to the comment #4).

The last sentence in section 3.3.3 has been revised as follows.

P10, L27–30

Because in situ $\varepsilon_{photolysis}$ used in the model calculation is nearly the same between low and high latitudes (Fig. S6c), this agreement supports the inference that the major causes of the difference are transport and mixing, which was previously suggested by observations in the high latitudes (Park et al., 2004) and by 1D or 3D model studies (McLinden et al., 2003; Morgan et al., 2003).

6-15) Page 8, Line 1-4: "There is no bending or curved structure apparent in the Rayleigh diagram, which suggests that chemical processes are not variable, although the small fluctuation in the lower left region in Fig. 5a will be discussed later." First, "chemical processes are not variable" will not be clear to many readers as to what the authors are trying to communicate regarding the variability of the processes involved. Second, in addition to my not agreeing that the lower left region of Fig 5a means something about O(1D) chemistry dominating photolysis (see above major comments), I also find this statement misleading: Chemical processes can vary throughout the stratosphere but the isotopic composition of N2O – being a

long-lived tracer – is not necessarily going to retain a 'memory' of this since air with many different transport histories (and chemical histories) are mixed together; in fact, N2O destruction is dominated by photolysis in upper stratosphere, and then the remaining N2O is redistributed from the upper stratospheric sink region. So I don't think the authors can make the inference the other way around – that if there is no bending or curvature in the Rayleigh plot, that that means that chemistry and relative chemical time scales do not vary throughout the stratosphere.

We agree that the sentences are not clear and misleading. They have been revised as follows.

P9, L9–14

> Especially in Fig. 5b, almost all data show a compact linear relation without bending or curved structure apparent in the Rayleigh diagram. The slope, which corresponds to the ratio of $\varepsilon$ values, is very close to the one expected for photolysis. This confirms that photochemical decomposition of $N_2O$ is mainly caused by photolysis, although the small fluctuation in the lower left region in Fig. 5a will be discussed later.

6-16) If the authors do wind up using their data to support a large relative role for photooxidation in the lower stratosphere, then they need to more explicitly differentiate their comments on Page 9, line 25 that "transport and mixing" are discussed as the difference between the epsilon values at different the latitudes, from those on Page 10, Line 4 that photooxidation is cited as having a major influence in the lower stratosphere (for which it is the ratios of epsilon values are assumed to be independent of transport of mixing); this is confusing without clarification (for which background is lacking in general for this "ratio of ratio" approach).

Sentences in the last half of section 3.4 have been revised. See our response to the comment #5(b).

6-17) Throughout: "Transport" is the term used to describe air mass or tracer transport (while transportation is buses, cars, airplanes, etc.)

"transportation" has been replaced with "transport", as suggested.

6-18) By the way, the so-called "transport barrier" is not really a barrier – it just looks like one in satellite views of tracers like the aerosol from the Mt Pinatubo eruption! It 'looks' and 'acts' like a barrier because, between 22 and 28 km, the vertical ascent rates in the tropics are much faster than quasi-horizontal transport rates out of the tropics to the midlatitudes, which in turn are much much faster then quasi-horizontal rates from the midlatitudes into the tropics (and contrasting with the rate of transport and mixing into the tropics below 22 km, which your data here, and many other types of tracer studies, are completely consistent with!). Alan Plumb discusses this quite a bit, as 'barrier' is a bit of a misnomer when one is contemplating the dynamics and transport.

The sentence has been revised as follows.

P9, L26–28

> Because the vertical ascent rate in the tropics is much faster than quasi-horizontal transport, there is

an apparent transport barrier between the tropics and extratropics (Plumb, 2007). Nevertheless, entrainment of air mass Yi ….

---

## Author Comment (AC3) · 21 Aug 2017

Reply to co-editor

Co-editor's comment is typed in blue, authors' response is typed in black.

I have one more scientific comment that you may want to consider already, although I do not require any changes now. It may also be addressed in the public discussion later. It relates to the value of your 2-D mixing model

1) Is it realistic to assume an end member with only 5 ppb N2O in the tropical region?

It is true that the mixing ratio we observed at about 30 km height is much higher than 5 ppb. But at higher altitudes in the middle/higher stratosphere (40–50 km), it would approach to 5 ppb. This can be seen in the 3-D model simulation results (Fig. 8a).

2) Conceptually, you develop an additional model to explain the shape of the curve in mid latitudes, which can already be successfully explained by the 1D model. But your model does not provide any evidence that the epsilon values in the equatorial mid stratosphere, where you present the new data, are so much higher. So in the present text I am not really convinced about the additional value of this simple model. The 3D model result on the other hand is very valuable!

The referee #2 pointed out that the discussion with the 1-D model is misleading and even incorrect. We decided to delete the 1-D model part and leave the discussion with the conceptual 2-D model according to the referee's comment.

---

## Author Response (AR2)

Reply to the referee #2

Referee's comment is typed in blue, authors' response is typed in black, and the change in the revised text is highlighted with red.

The authors have made a number of improvements to the original manuscript, and they have included additional details and raw data. In my opinion, the manuscript is now publishable without further review on my part. However, I do offer some additional comments for the authors to consider if they wish to clarify and revise any of the following remaining issues.

We appreciate the referee for his/her further constructive comments. We have revised the manuscript according to his/her suggestions.

Abstract: P1, L32-33: I suggest adding the very important clarification in caps: "These results FROM THE DEEP TROPICS suggest the following."

Done.

Abstract: P1, L33-35 The authors state: "The time scale of horizontal mixing in the tropical middle stratosphere is sufficiently large for in-situ photolysis of N2O, mainly because of strong upwelling and transport barrier between the tropics and extratropics." What does this sentence mean? Taken literally, it doesn't make any sense to me and is oddly phrased. I am guessing that authors are trying to say that the epsilon values in the Rayleigh analysis for the N2O isotopocules are as large (or almost as large) as those for photolysis in a closed system because air in the middle tropical stratosphere is relatively isolated from the mixing in of older, photochemically-aged air from the extratropics (which serves to decrease epsilon values from the Rayleigh limit). If I were to just try to rewrite their sentence (which I think would not be as clear), then it would be more accurate to say something like: "The time scale for quasi-horizontal mixing between tropical and midlatitude air in the tropical middle stratosphere is sufficiently slow relative to the tropical upwelling rate that isotope fractionation approaches the Rayleigh limit for N2O photolysis."

There is a similar problematic wording in the Conclusions (see below).

Revised as suggested.

P3, L32: The authors state that (Kaiser et al., 2002a) "estimated that a much larger fraction than 10% is removed by photooxidation at least in the lower stratosphere" Can the authors please give a number for the larger fraction that Kaiser et al suggested? How much larger than 10%?

We have revised the sentence as follows.

> …they estimated that a much larger fraction (up to 100%) is removed by photooxidation at least in the lower stratosphere (N2O mixing raitos > 300 nmol mol-1) (Kaiser et al., 2006).

P3 L33: The authors state that "similar but a little simplified analyses by Park et al. (2004) and Toyoda et al. (2004)" were performed. Can the authors please be more descriptive in what they mean by 'but a little

simplified relative to Kaiser et al"?

Since we described this in section 3.4, we have just added a reference to section 3.4 to the sentence.

P6, L4: The authors say that they used CO2 mixing ratios to estimate the mean age of air of an air sample and cite Engel 2009 for the measurements, but is this true for all flights (i.e., that CO2 mixing ratios were measured for all samples by Engel et al. (2009))? Or perhaps they just mean to state how a mean age can be obtained from CO2 mixing ratios? This is unclear and should be clarified in either case.

We have revised the sentence (changed the position of "Engel et al., 2009") to mean to state how a mean age can be obtained from CO2 mixing ratios.

P9, L11-12: The authors state: "The slope, which corresponds to the ratio of epsilon values, is very close to the one expected for photolysis. This confirms that photochemical decomposition of N2O is mainly caused by photolysis (Minshwaner et al., 1993)…" I think most stratospheric chemists will think it is a stretch that this isotope data needs to be used to confirm what has been known for some time. In other words, I don't believe the fact that the majority of N2O is destroyed by photolysis needs to be confirmed in 2017. This should be reworded so that it doesn't seem like this is an important, new, or controversial finding.

We have revised the sentence as follows.

> This agrees with the fact that photochemical decomposition of $N_2O$ is mainly caused by photolysis (Minshwaner et al., 1993), …

P9, L27-28 The authors state here: "Because the vertical ascent rate in the tropics is much faster than quasi-horizontal transport, there is an apparent transport barrier between the tropics and extratropics (Plumb, 2007)…" Sort of… It would be better to take out the 'much' from "much faster" and then qualify by saying "Because the vertical ascent rate in the tropics is faster than quasi-horizontal transport out to the extratropics and much faster than the quasi-horizontal transport of extratopical air into the tropics, there is an apparent transport barrier between the tropics and extratropics…" On the other hand, what one really wants to get across for this isotope study is that the tropics are relatively isolated than the rest of the stratosphere – especially the middle tropical stratosphere since little extratropical air is mixed back in (i.e., the vertical ascent rates and the entrainment of tropical air out to midlatitudes are both much faster than the transport of older air back into the tropics…).

The sentence has been revised as suggested.

P11, L11: The authors state: "Low values are obtained near the TTL over the Equator." Note that the TTL is usually tropospheric air, not stratospheric. So are these samples in the troposphere (below the tropopause) or above the tropopause in the stratosphere ?

The sentence has been revised as follows.

> … low values are obtained just above the TTL over the Equator (EQP, $z$ = 20 km) just as they are at

other latitudes.

P11, L11: The authors state: "This result confirms the indication by Kaiser et al. (2006) that the photooxidation sink has a much larger fraction than 10% in the lower stratosphere." As I noted earlier, please give a number for 'much larger'? And 'confirms'? I would not agree yet that the analysis presented here confirms that at all. So I agree with the authors, who state below, that this will require more study. But, in light of this, perhaps they could use a word other than 'confirms' which carries a strong meaning scientifically, which is a bar I do not think they have met yet. It's fine to publish this idea, and the case they present, but I think "confirm" is too strong a word to use just yet.

The sentence has been revised as follows. "Number for 'much larger' " has been added in p. 3 as described above.

This result is in accordance with the indication by Kaiser et al. (2006) ….

P11, L13-15: Figure S6: It is an important new figure component that the authors now provide in this revised manuscript the total loss rate for N2O in relevant regions in the stratosphere. I still believe they are not interpreting it nor their data in as rigorous a way as they could, but, as they note, they leave further work up to future studies. However, in the authors' response to review they note "[we] deleted a part of the plots of tropics and all the plots of February Arctic region so that the plots show the parameters in the region where photochemical N2O loss really occurs." I do see tropical parameters in their Fig S6, so I am wondering what 'part' was deleted. I think it is very important to include a complete Fig S6 so that readers can evaluate and follow up on the arguments that the authors are making here.

The part of plots deleted from the tropical profiles are those below 17 km, which was the lowest height of cold point tropopause during our observations.

Conclusions:
P 11, L28-29 "Unlike other region of the stratosphere, enrichment factors for isotopocules in the middle equatorial stratosphere (25–30 km, or [N2O] < 260 nmol mol-1) agreed with those obtained with laboratory photolysis experiments, suggesting that the isotopocule ratios are determined mainly by photolysis because of weak vertical or horizontal mixing in the tropical upwelling…" As I noted in a comment to the authors above, the wording seems to mix up cause and effect. The middle tropical stratosphere is relatively isolated; the lack of transport of air of many different transport time histories makes the isotope fractionation appear almost Rayleigh (as in an -- almost -- closed system). The specific arguments about weak vertical and horizontal mixing are not especially accurate, nor are they really needed.

The sentence has been revised as follows.

Unlike other region of the stratosphere, enrichment factors for isotopocules in the middle equatorial stratosphere (25–30 km, or $[N_2O] < 260$ nmol mol$^{-1}$) agreed with those obtained with laboratory photolysis experiments because the middle tropical stratosphere is relatively isolated than the rest of the stratosphere.

P12, L3-5: The authors state: "Further observations of temporal variations and comparison with ACTM simulation will be needed to examine the change in the meridional circulation…" What change in the meridional circulation? This is the first time the authors have brought this up as a motivation or next step. There is a vast literature on this hot topic but no context is presented here at all, and no references are given. If the authors wish to motivate that this is something that N2O isotopocules could provide insight into, they should provide at least a sentence or two of context as well as some references, and a more complete argument of how N2O isotopocules could help.

The sentence has been revised as follows.

> Further observations of temporal variations and comparison with ACTM simulation will be needed to obtain the quantitative estimate of the importance of photooxidation pathway and to examine the possible change in the meridional circulation predicted by model simulations (Li et al., 2008; McLandress and Shepherd, 2009) because such a change might affect $\varepsilon$ values of stratospheric $N_2O$ by perturbing mixing of air with different transport time histories.

Even more minor comments:

Polishing of the English usage of definite (the's) and indefinite (a's) articles could be helpful for readers.

The first version of our manuscript had been edited by a commercial English editing service. We have checked the revised sentences carefully and tried our best to polish the English.

Pg2,L24; "transportation" should be replaced with "transport"

We forgot to correct this in the previous revision, thank you.

Pg 3, L8-9: "Although these observations are limited to ca. 20 km altitude, vertical profiles can be obtained for horizontally wide areas such as Arctic polar vortexes (pg 3, L8-9)." It is unclear what is meant by this nor why this is an important distinction.

We intended to mean that aircraft observations have a limitation of altitude and have the advantage in obtaining horizontal distributions of trace gasses and their isotopocules. We have revised the sentence as follows.

> Although these observations are limited to up to ca. 20 km altitude, mean vertical profiles can be obtained from horizontally wide areas such as Arctic polar vortexes.

P7, L25-28: "The height of the tropical tropopause layer (TTL) was 14-18.5 km (Fueglistaler et al., 2009). I presume since the measurements were made in 2015, then the authors are using Fueglistaler to mean "typically," and what the TTL is rather than a reference that has the actual tropopause measurements for these flights in it. So saying 'typically' and putting an "e.g.," in the reference could clarify this.

The sentence has been revised as follows, with correction of table number.

The height of the tropical tropopause layer (TTL) is typically 14–18.5 km (e.g., Fueglistaler et al., 2009), whereas the tropopause height was 12–16 km over Japan and 9 or 10 km over Sweden and Antarctica (Table S2).

P8, L19: The authors state: "The equatorial values of epsilon almost coincide with those of photolysis…" If the authors mean "almost as large as" it would be more clear to state it that way (since it gives the direction, and what is expected); if it is noisy, then I suppose "almost coincide" would be ok.

Revised as suggested.

P8, L21-22: The authors state "latitudinal and year-to-year or seasonal variation are slight compared to those of the middle stratosphere in the lower stratosphere": I had to read this several times, as the order is confusing. I think the authors mean mean: "latitudinal and year-to-year or seasonal variation are slight IN THE LOWER STRATOSPHERE compared to VARIATIONS IN the middle stratosphere"

Revised as suggested.

P8, L22-25 The authors state: "Although the similar latitudinal and altitudinal dependence of epsilon has been reported previously for the latitudes ranging from 18°N to 89°N (Park et al., 2004; Kaiser et al., 2006), our equatorial data showed that the change in epsilon at altitude with higher N2O mixing ratio and the epsilon value is exactly what would be expected during the N2O photolysis." I have several suggestions to make this sentence more clear. First, putting "show" in present tense (instead of past tense "showed") clarifies that the authors are referring to the new deep tropical data presented here. In addition, the very last part of the sentence I think is a preview of what will be discussed in the next section, not a statement that needs to be supported here.

The sentence has been divided in two and revised as follows.

…, our equatorial data show that the change in $\varepsilon$ at altitude with higher $N_2O$ mixing ratio. The $\varepsilon$ value is exactly what would be expected during the $N_2O$ photolysis as discussed below.

P8, L29: We NOW discuss causes (not 'then').

Done.

[revised manuscript text omitted]